# Biomimetic nanovaccine-mediated multi-valent IL-15 self-transpresentation (MIST) for potent and safe cancer immunotherapy

Kaiyuan Wang [1,2,9], Xuanbo Zhang[1,2,9], Hao Ye [1,3,9], Xia Wang[4], Zhijin Fan [5], Qi Lu[1], Songhao Li[1], Jian Zhao[1], Shunzhe Zheng[1], Zhonggui He [1] ✉, Qianqian Ni[2,6,7] ✉, Xiaoyuan Chen [2,6,7,8] ✉ & Jin Sun [1] ✉

Cytokine therapy, involving interleukin-15 (IL-15), is a promising strategy for cancer immunotherapy. However, clinical application has been limited due to severe toxicity and the relatively low immune response rate, caused by wide distribution of cytokine receptors, systemic immune activation and short half-life of IL-15. Here we show that a biomimetic nanovaccine, developed to co-deliver IL-15 and an antigen/major histocompatibility complex (MHC) selectively targets IL-15 to antigen-specific cytotoxic T lymphocytes (CTL), thereby reducing off-target toxicity. The biomimetic nanovaccine is composed of cytomembrane vesicles, derived from genetically engineered dendritic cells (DC), onto which IL-15/IL-15 receptor α (IL-15Rα), tumor-associated antigenic (TAA) peptide/MHC-I, and relevant costimulatory molecules are simulta-neously anchored. We demonstrate that, in contrast to conventional IL-15 therapy, the biomimetic nanovaccine with multivalent IL-15 self-transpre-sentation (biNV-IL-15) prolonged blood circulation of the cytokine with an 8.2-fold longer half-life than free IL-15 and improved the therapeutic window. This dual targeting strategy allows for spatiotemporal manipulation of therapeutic T cells, elicits broad spectrum antigen-specific T cell responses, and promotes cures in multiple syngeneic tumor models with minimal systemic side effects.

Cytokine therapy is one of the first US Food and Drug Administration (FDA)-approved immunotherapeutic approaches[1]. Specifically, the antitumor cytokine interferon-alpha (IFN-α) and T cell growth factor interleukin-2 (IL-2) have been approved to treat many kinds of advanced tumors in 1986 and 1992, respectively[2,3]. The advances of IFN-α and IL-2 therapies confirmed that polarizing the immunologic balance to proinflammatory phenotype could delay the progression of cancers[4]. Emerging evidence shows that IL-15 plays a critical role in the homeostasis, initiation, and development of anticancer immunity[5]. IL-15 is a membrane-associated growth factor to elicit biological

[1]Department of Pharmaceutics, Wuya College of Innovation, Shenyang Pharmaceutical University, 103 Wenhua Road, Shenyang, Liaoning 110016, P. R. China. [2]Departments of Diagnostic Radiology, Surgery, Chemical and Biomolecular Engineering, and Biomedical Engineering, Yong Loo Lin School of Medicine and College of Design and Engineering, National University of Singapore, Singapore 119074, Singapore. [3]Multi-Scale Robotics Lab (MSRL), Institute of Robotics & Intelligent Systems (IRIS), ETH Zurich, Zurich 8092, Switzerland. [4]School of Pharmacy, Shenyang Pharmaceutical University, Shenyang, Liaoning 110016, China. [5]School of Medicine, South China University of Technology, Guangzhou 510006, P.R. China. [6]Clinical Imaging Research Centre, Centre for Transla-tional Medicine, Yong Loo Lin School of Medicine, National University of Singapore, Singapore 117599, Singapore. [7]Nanomedicine Translational Research Program, Yong Loo Lin School of Medicine, National University of Singapore, Singapore 117597, Singapore. [8]Institute of Molecular and Cell Biology, Agency for Science, Technology, and Research (A*STAR), 61 Biopolis Drive, Proteos, Singapore 138673, Singapore. [9]These authors contributed equally: Kaiyuan Wang, Xuanbo Zhang, Hao Ye. ✉e-mail: hezhgui_student@aliyun.com; qqian.ni@nus.edu.sg; chen.shawn@nus.edu.sg; sunjin@syphu.edu.cn

responses through binding to a high-affinity heterotrimeric receptor complex composed of IL-15 receptor α (IL-15Rα) and IL-15Rβ/γ[6]. IL-15 binds to IL-15Rα on the membrane of antigen-presenting cells (APC), followed by transpresentation to IL-15Rβ/γ expressed on the neighboring T cells[7]. Indeed, IL-15 and IL-2 activate similar signaling transduction cascades, e.g., through binding to the common IL-2/IL-15Rβ/γ chains[8]. Recent data reveal that IL-2 and IL-15 differ in the roles of lymphocyte activation. IL-15 supports the generation and survival of memory T cells, while IL-2 promotes the expansion of all T cell subsets including effector T cells and immunosuppressive regulatory T cells (Treg)[9–11]. Although promising, clinical translation of IL-15 therapies is limited owing to either suboptimal efficacy or dose-limited toxicity[12,13]. The weak interaction with tumor-specific T cells of IL-15 raises the risks of nontumor-specific immune activation, leading to systemic side effects and a deficiency of tumor suppression[14].

In recent years, diverse efforts have been devoted to overcoming the off-target and non-responder issues of cytokines through nanomedicine approaches for efficient cytokine delivery[15–17]. Despite the favorable outcomes, current cytokine delivery strategies still suffer from premature leakage before reaching target sites and undesirable induction of nontumor-specific cytotoxic T lymphocytes (CTL)[18]. Accordingly, targeted delivery of IL-15 to tumor-specific T cells is of great essence for preventing systemic side effects and augmenting antitumor immunity of IL-15 therapy[19]. In addition, the mode of IL-15 presentation (configuration) is critical for optimal immune stimulation[20]. Multivalency depicts the simultaneous interactions between multivalent ligands and receptors, which may improve specificity and avidity to IL-15 transpresentation[21,22]. Advances in nanomedicine technologies have opened the possibility for controlled delivery of multivalent IL-15 with T cell targeting ligands or therapeutics to minimize off-target delivery of IL-15 to nontumor-specific T cells and induce potent antitumor immunity through multivalent interactions. Among the numerous nanomedicine strategies, biomimetic nanotechnology has been studied for a long time[23–26]. Compared to conventional approaches, biomimetic nanotechnology enables immunoregulation via prolonged circulation, excellent biocompatibility, minimal toxicity, and high targeting capability[27–31].

As a proof of concept, we design a biomimetic nanovaccine by engineering dendritic cell (DC)-derived cytomembrane vesicles to enable spatiotemporally synchronous cytokine delivery (biNV-IL-15). The defined biomimetic nanovaccine features multivalent IL-15 self-transpresentation, directional presentation of tumor antigen epitopes via major histocompatibility complex (MHC) molecule, and co-stimulatory molecules anchoring through a programmable procedure. We find that the biomimetic nanovaccine is endowed with a nanoscale-sized structure, prolonged circulation half-life and retention in lymphatic organs, improved therapeutic window, and stronger antitumor potency. In contrast to conventional IL-15 therapy, which requires binding to IL-15Rα on APCs, the biomimetic nanovaccine can directly mediate IL-15 transpresentation to CD8+ T cells in a multivalent manner to elicit robust T cell activation, which is herein termed as "multivalent IL-15 self-transpresentation (MIST)". Systemic administration of biNV-IL-15 achieves targeted delivery of IL-15 to tumor-specific T cells, diminishes non-specific systemic immune stimulation, and maximizes CTL activation to drive tumor killing and immunological memory. This strategy displays promise as a safe, effective, and generalizable platform technology for the clinical development of IL-15 immunotherapy.

## Results

### Preparation and characterization of biNV-IL-15

As shown in Fig. 1a, we developed a biomimetic nanovaccine (biNV) via the engineering of dendritic cell (DC)-derived cytomembrane vesicles that efficiently delivered IL-15 to tumor-specific T cells for potent and safe cancer immunotherapy. DC vesicles are expected to precisely and effectively deliver IL-15 to tumor-specific T cells: (1) the molecular components of biNV are similar to living DCs, including surface expression of major histocompatibility complex (MHC) I/antigenic peptide complexes and costimulatory molecules[32], (2) the cell-free biNV minimizes the cellular therapy-related risks such as in vivo replication, (3) biNV displays extensive distribution in lymph nodes due to suitable size (110 nm) and abundant surface membrane receptors. Additionally, multivalent IL-15/IL-15Rα were expressed on the membrane of biNV through genetic engineering to generate the final product of biNV-IL-15. The resulting biNV-IL-15 could present both costimulatory molecules (e.g., B7-1 and B7-2) and tumor antigens to naive tumor-specific T cells. It is important that multivalent interactions between IL-15/IL-15Rα (biNV-IL-15) and IL-15Rβ/γ (T cells) lead to high specificity and strong binding affinity for biomimetic nanovaccine-mediated IL-15 self-transpresentation. Notably, the interaction between T cell receptor (TCR) and biNV could promote the targeted delivery of IL-15 to tumor-specific T cells, which subsequently facilitates synergistic antitumor T cell responses. Additionally, tumor-specific cytotoxic T lymphocytes (CTL) activated by biNV-IL-15 could produce proinflammatory cytokines, including tumor necrosis factor (TNF)-α and interferon (IFN)-γ, which mediate phenotypic polarization of M2-type tumor-associated macrophages (TAM) to antitumoral M1-type TAMs, in turn, impede the immunosuppressive effects of regulatory T cells (Treg) within the tumor microenvironment (TME) (Fig. 1b)[33]. Moreover, biNV-IL-15 would achieve spatiotemporally synchronous delivery of multivalent IL-15 and tumor antigen to the individual CD8+ T cells, thereby enabling stronger capability of antigen-specific T cell activation than co-administration of mixed IL-15 and biNV (IL-15+biNV), where T cells mainly interact with either IL-15 or biNV separately (Fig. 1c).

We first explored the validity and feasibility of preparing genetically engineered biNV-IL-15 for multivalent IL-15 self-transpresentation. Presenting endogenous tumor antigens is an effective way to elicit cancer-specific immune responses and could simultaneously induce durable immune memory. However, most endogenous tumor antigens remain unclear for many cancers. Heat shock protein 70 (HSP70) has been demonstrated to be a chaperone of some polypeptides to produce tumor antigens, which could be presented by DCs to activate CD8+ T cell-mediated immune response[34]. In our study, biNV was generated from DCs pulsed with endogenous HSP70-chaperoned polypeptides (HCP). The HSP70 expression in normal tissue and tumor tissue was analyzed using immunohistochemistry (IHC). As shown in Fig. 2c, HSP70 was overexpressed in 4T1 and CT26 tumor tissues, suggesting that abundant tumor antigens existed in the chaperoned polypeptides as compared with normal tissues. Tumor tissues were homogenized to isolate HCP through the immunoprecipitation technique (approximately 50 μg of HCP was obtained per gram of tumor tissue). As shown in Fig. 2d, e, the average diameter was around 10 nm from dynamic light scattering (DLS), and the molecular weight of HCP was about 50-70 kDa indicated by coomassie blue protein staining. To develop highly efficient membrane-bound IL-15 anchored DC vesicles with tumor antigen presentation ability (biNV-IL-15), bone marrow-derived DCs (BMDC) were transduced with a combination of mouse IL-15 and IL-15Rα encoded recombinant adenovirus (IL-15/IL-15Rα rAD) and then were pulsed with the endogenous HCP (DC-IL-15/HCP) (Fig. 2a). The maturation of DC-IL-15/HCP was confirmed by flow cytometric examination gated on CD11c+CD80+CD86+ (Supplementary Fig. 1). Afterwards, biNV-IL-15 was isolated through multistep density gradient ultracentrifugation (Fig. 2b)[35]. 39.27 ± 5.93 mg of DC vesicles can be harvested from 10^8 DCs. The uniform vesicular morphology of biNV-IL-15 was determined with the transmission electron microscope (TEM) and immunogold labeling was performed to visualize the surface distribution of MHC-I molecules (Fig. 2f). Meanwhile, nanoparticle tracking analysis

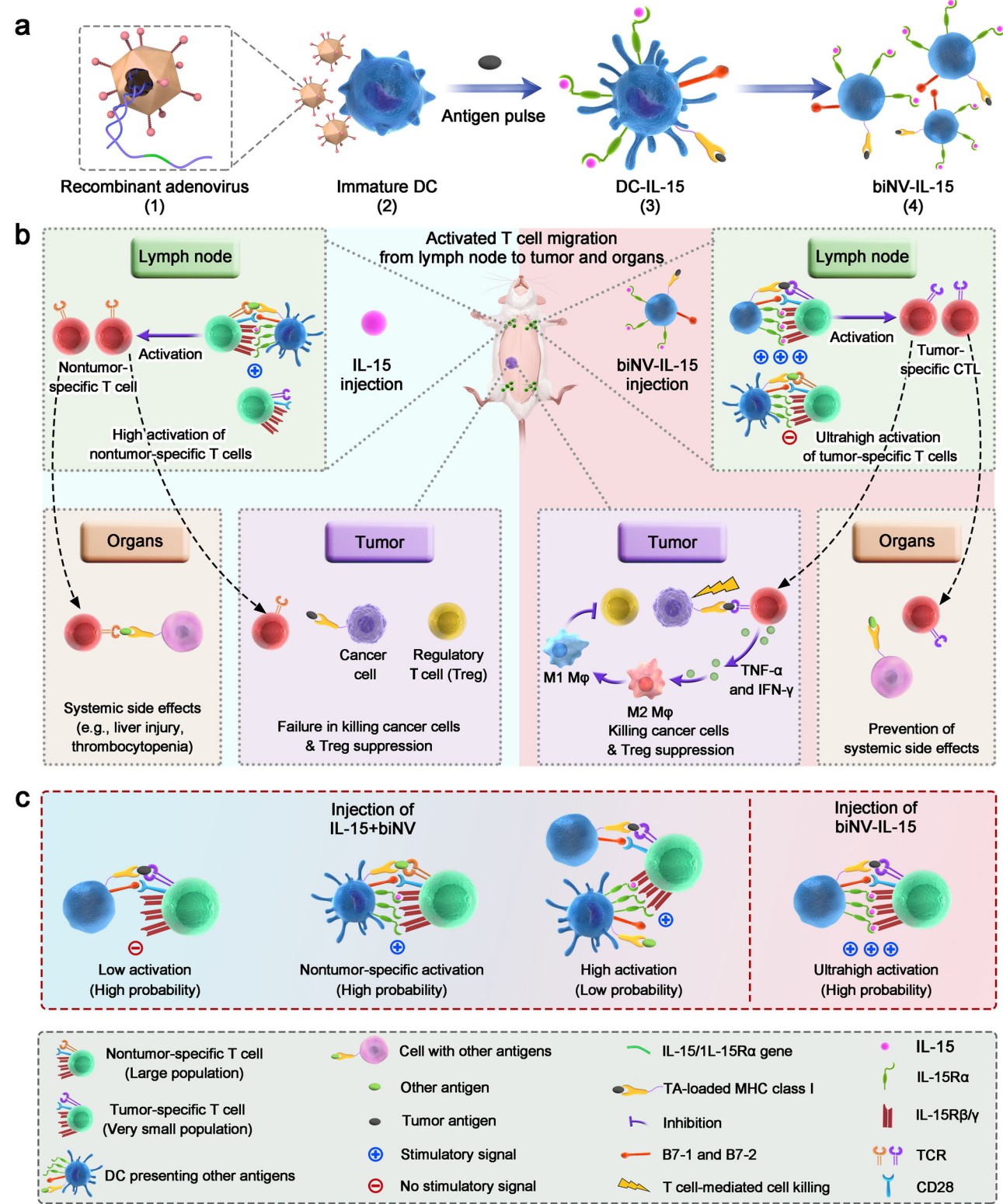

(NTA) showed that biNV-IL-15 had a mean diameter of 110 nm with narrow size distribution (Fig. 2g). DC vesicles could remain stable in PBS at 37 °C or 4 °C, and the diameter demonstrated negligible changes after cryopreservation at −80 °C for 60 d (Supplementary Fig. 2). To determine the co-localization of IL-15 and MHC-I, DCs transduced with GFP-tagged IL-15/IL-15Rα rAD were used for DC vesicle isolation and was then incubated with the anti-MHC-I antibody for further Alexa Fluor 594 conjugated secondary antibody labeling. Confocal laser scanning microscopy (CLSM) imaging

showed the co-localization of green and red fluorescence (Fig. 2h), indicating the successful co-expression of IL-15 and MHC-I on the surface of biNV-IL-15. B7-1/2 co-stimulatory molecules are known to participate in the antigen presentation process and promote immunity by triggering the CD28 receptors on T cells. The western blotting analysis also confirmed the expression of B7-1/2 on biNV-IL-15. Besides, we found the expression of intercellular adhesion molecule-1 (ICAM-I) and CC-chemokine receptor 7 (CCR7) on biNV-IL-15, which could facilitate the migration to lymph nodes (LNs)

**Fig. 1 | Biomimetic nanovaccine-mediated IL-15 self-transpresentation in a multivalent form can improve the efficacy and avoid the side effects of IL-15. a** Schematic representation describing the stepwise fabrication of biNV-IL-15. (1) Adenovirus vectors were genetically modified with IL-15/IL-15Rα gene. (2) Recombinant adenovirus transduced the BMDCs to express membrane-bound IL-15 on the cell surface. (3) Tumor antigen priming and DC maturation. (4) The isolation of biNV-IL-15. **b** Schematic representation depicting the hypothesis of the research: biNV-IL-15 induces the ultrahigh activation of tumor-specific CTLs. Conventional IL-15 treatment has a high probability of binding to nontumor-specific T cells with a large population in the lymph node, resulting in unsatisfactory therapeutic efficacy

and serious side effects. On the contrary, biNV-IL-15 would target tumor-specific T cells and simultaneously deliver multivalent IL-15 and tumor antigen to exclusively stimulate tumor-specific T cell activation, improving therapeutic efficacy and preventing side effects. **c** biNV-IL-15 treatment is more efficacious in tumor-specific T cell activation than IL-15 and biNV co-administration. The T cells have a high possibility to interact with either IL-15 or tumor antigen as well as a low possibility to interact with both IL-15 and tumor antigen through co-administration, leading to inefficient tumor-specific CTL production. By contrast, biNV-IL-15 has a high possibility to simultaneously provide IL-15 and tumor antigen to T cells, resulting in efficient tumor-specific CTL stimulation.

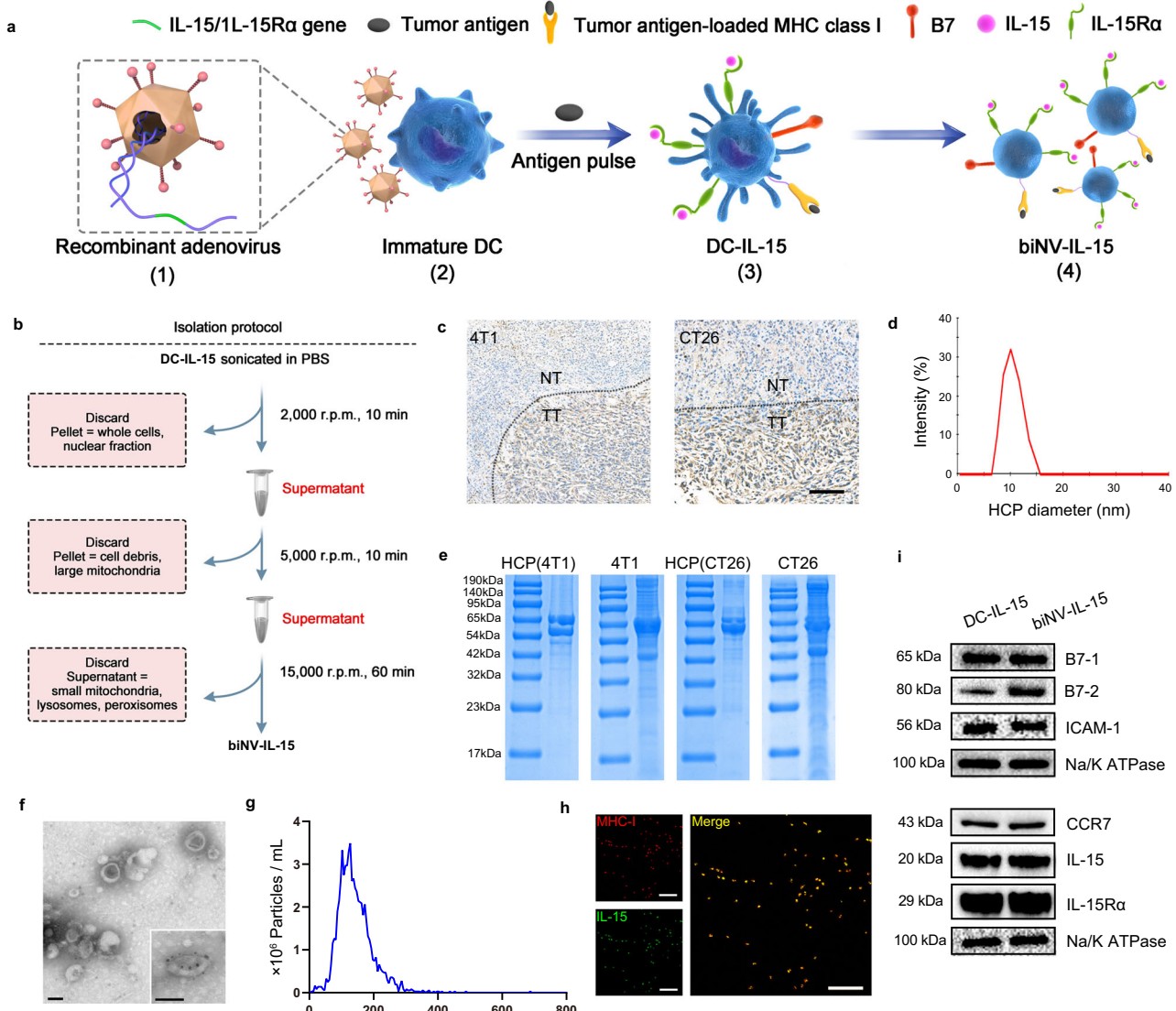

**Fig. 2 | Preparation and characterization of biNV-IL-15. a** Production of biNV-IL-15 isolated from recombinant adenovirus-transduced DCs. (1) Adenovirus vectors were genetically modified with IL-15/IL-15Rα gene. (2) Recombinant adenovirus transduced the BMDCs to express membrane-bound IL-15 on the cell surface. (3) Tumor antigen priming and DC maturation. (4) The isolation of biNV-IL-15. **b** Schematic representation of the production of biNV-IL-15. **c** The comparison of HSP70 expression between normal tissue (NT) and tumor tissue (TT) on 4T1 and CT26 tumors through immunohistochemistry. Scale bar = 100 μm. **d** The diameter

of extracted HCP measured by DLS. **e** SDS-PAGE protein analysis of the isolated HCP and 4T1, CT26 tumors. **f** TEM images of biNV-IL-15. Inset: immunogold labeling of MHC-I (10 nm gold particle). Scale bar = 100 nm. **g** NTA analysis of biNV-IL-15. **h** CLSM images of MHC-I (red) and IL-15 (green) on biNV-IL-15. Scale bar = 1 μm. **i** Western blotting analysis of B7-1, B7-2, ICAM-I, CCR7, IL-15, and IL-15Rα on biNV-IL-15. For panels **c, e, f, h**, experiment was repeated three times independently with similar results. For panel **i**, experiment was repeated twice independently with similar results. Source data underlying panel **g, i** are provided as a Source Data file.

(Fig. 2i)[36,37]. Moreover, an enzyme-linked immunosorbent assay (ELISA) was performed to quantify the IL-15 expression. Compared with IL-15 monotransduction, the DC vesicles manifested prominent improvement in membrane-bound IL-15 expression after IL-15/IL-

15Rα cotransduction[38]. The binding efficiency could reach about 5.31 μg IL-15 per 100 μg DC vesicles (Supplementary Fig. 3). The above results indicated that the approach we described for biNV-IL-15 preparation was valid and feasible.

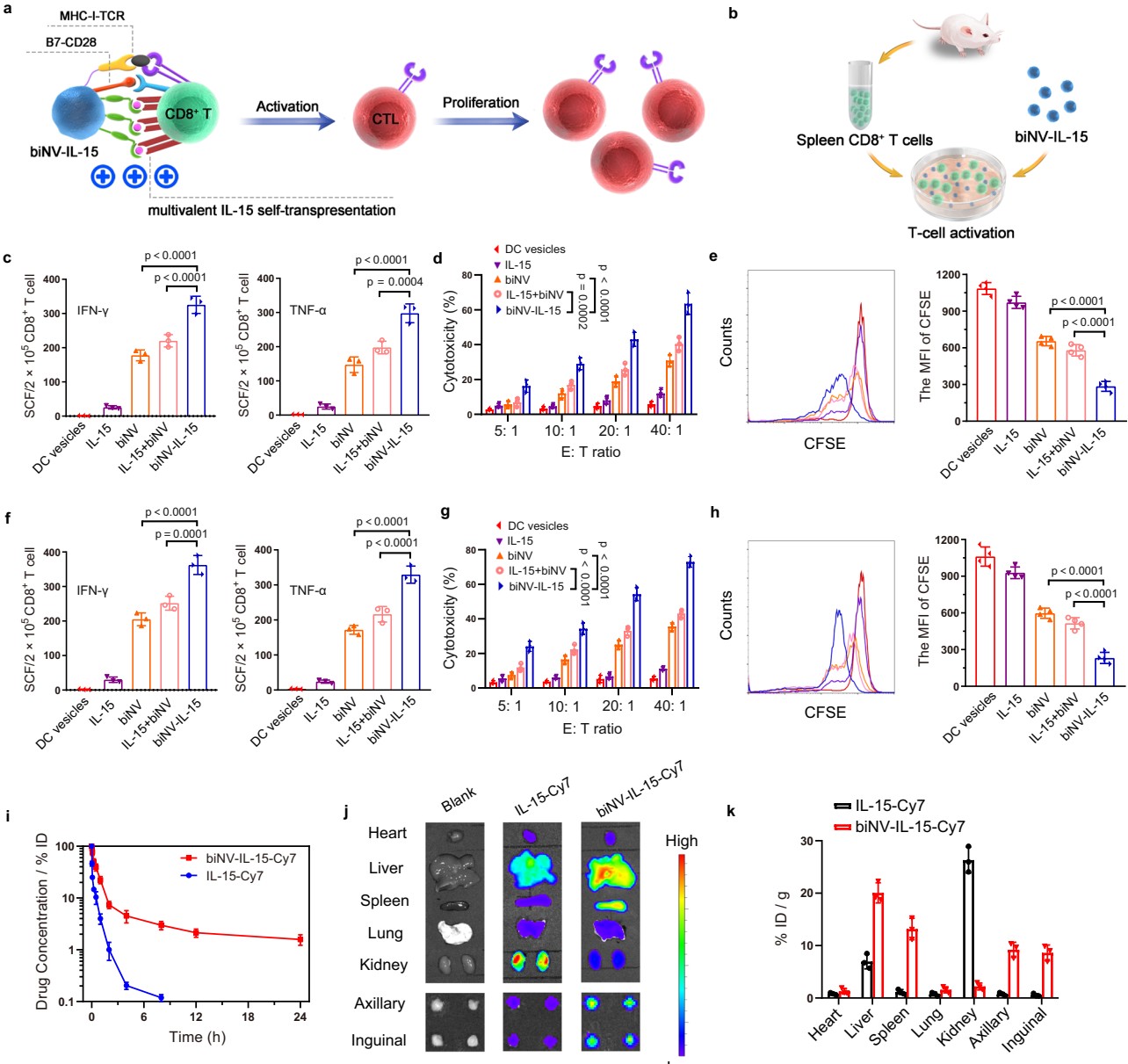

**Fig. 3 | biNV-IL-15 for multivalent IL-15 self-transpresentation (MIST) and secondary lymphoid organ accumulation. a** Schematic depicting biNV-IL-15 to activate CD8+ T cells and stimulate their proliferation. **b** The direct activation of splenic CD8+ T cells harvested from BALB/c mice, $2 \times 10^5$ cells were coincubated with various formulations (DC vesicles, IL-15, biNV, IL-15+biNV, and biNV-IL-15) for 7 d. HCP isolated from 4T1 (**c–e**) and CT26 (**f–h**) tumor tissue was used for DC priming and DC vesicle isolation for T cell activation, respectively. **c, f** ELISPOT analysis of IFN-γ and TNF-α spot-forming cells (*n* = 3/group). Cytolysis rate against 4T1 (**d**) and CT26 (**g**) cancer cells with T cells after various activations. The cytolysis efficiency was measured through LDH-releasing cytotoxic assay with effector-to-target cell (E:T) ratios of 5:1, 10:1, 20:1, and 40:1 (*n* = 3/group). The statistical analysis was performed at the E:T ratio of 40:1. **e, h** The flow cytometric detection and mean

fluorescence intensity (MFI) of T lymphocyte proliferation with CFSE staining after various treatments (*n* = 4/group). **i** Pharmacokinetic profiles of Cy7-labeled IL-15 and biNV-IL-15 by assessing the fluorescence of Cy7 in blood samples at predetermined time points (*n* = 3/group). **j** Fluorescence imaging and **k** quantitative analysis of isolated organs, axillary and inguinal LNs after intravenous injection of Cy7-labeled IL-15 and biNV-IL-15 (*n* = 3/group). Data represent the mean ± s.d. The *p* values of biNV-IL-15 to biNV in panels **c–h** are <0.0001. And the *p* values of biNV-IL-15 to IL-15+biNV in panels **c–h** are <0.0001, 0.0004, 0.0002, <0.0001, 0.0001, <0.0001, <0.0001, and <0.0001, respectively. Statistical significance was calculated through one-way ANOVA using a Tukey post-hoc test. Source data underlying panels **c–i**, **k** are provided as a Source Data file.

## The direct activation of naive CD8+ T cells

We next explored the naive CD8+ T cell priming by biNV-IL-15 (Fig. 3a). The splenic CD8+ T cells were harvested and incubated with different formulations (DC vesicles, IL-15, biNV, IL-15+biNV, and biNV-IL-15) (Fig. 3b)[39]. HCP worked as tumor antigen was separately extracted from 4T1 or CT26 tumor tissue to fabricate biNV-IL-15, and DC vesicles were prepared from BMDCs without any treatment through multistep density gradient ultracentrifugation. The elicitation of T cell responses

was evaluated by detecting IFN-γ and TNF-α spot-forming cells through enzyme-linked immune absorbent spot (ELISPOT) assay[40]. The cytolysis efficiency of T cells against 4T1 or CT26 cancer cells was detected via lactate dehydrogenase (LDH)-releasing cytotoxic assay with effector-to-target cell (E:T) ratios ranging from 5:1, 10:1, 20:1, to 40:1. As shown in Fig. 3c, f, CD8+ T cells co-cultured with DC vesicles or IL-15 showed weak activation due to the failure in antigen presentation and IL-15 transpresentation. By contrast, biNV could stimulate CD8+

T cells to generate increased levels of IFN-γ and TNF-α, leading to a higher rate of T cell-mediated cytolysis against 4T1 and CT26 cancer cells (Fig. 3d, g). These results imply that DCs endowed biNV with intact surface functional proteins, which could mediate direct antigen presentation to naive T cells in vitro. In comparison with the IL-15+biNV mixture, biNV-IL-15 exhibited evident enhancement in IFN-γ and TNF-α production, resulting in above 70% killing efficiency with an E:T ratio of 40:1. These results proved the excellent CD8$^+$ T cell priming by biomimetic nanovaccine-mediated multivalent IL-15 self-transpresentation. Furthermore, the flow cytometric quantification of carboxyfluorescein diacetate succinimidyl ester (CFSE) staining indicated that biNV-IL-15 led to potent activation of CD8$^+$ T lymphocytes and increased their expansion compared with the other groups (Fig. 3e, h). These data confirmed the notable direct T cell activating capability of biomimetic nanovaccine-mediated IL-15 self-transpresentation.

## Pharmacokinetic and biodistribution study of Cy7-labeled DC vesicles

Successful IL-15 self-transpresentation in vivo relies on its efficient delivery to lymphatic organs. However, rapid systemic clearance and insufficient secondary lymphoid organ retention remain to be the major obstacles to the clinical application of IL-15. In our study, we labeled IL-15 and biNV-IL-15 with cyanine-7 (Cy-7)[41] and investigated their pharmacokinetic and biodistribution properties after intravenous administration (i.v.)[42]. As shown in Fig. 3i, IL-15-Cy7 was rapidly eliminated from blood circulation with a half-life ($t_{1/2}$) of 0.69 h. Encouragingly, biNV-IL-15-Cy7 displayed prolonged circulation time with $t_{1/2}$ around 5.66 h, which is much longer than that of IL-15-Cy7. The biNV-IL-15-Cy7 showed approximately 6.6-fold elevation in the area under the curve (AUC) compared with free IL-15-Cy7 and hence implied enhanced immune organ accumulation. We also tracked the biodistribution of biNV-IL-15-Cy7 and IL-15-Cy7 through ex vivo fluorescence imaging and measuring fluorescence of tissue lysate at 6 h post-administration. The axillary and inguinal LNs in biNV-IL-15-Cy7 treated mice exhibited 12.3- and 15.3-fold higher drug accumulation than those in IL-15-Cy7 treated mice. Meanwhile, biNV-IL-15-Cy7 enabled efficient delivery to the spleen, with an 11.2-fold higher fluorescence signal detected as compared with IL-15-Cy7 (Fig. 3j, k), suggesting that the retained lymphoid homing molecules on biNV could mediate the LN and spleen tropism of biNV-IL-15[43].

## The improved therapeutic window of biNV-IL-15 therapy

We next investigated the in vivo toxicity of biNV-IL-15. Healthy mice were treated with IL-15 and biNV-IL-15 with gradient dosing schemes (Fig. 4a)[44]. Significant weight loss was observed after four times injections of IL-15 with a single treatment dose above 7.5 μg, which ultimately would lead to lethal immunotoxicity (Fig. 4b). By contrast, no obvious toxicity appeared across the treatment of biNV-IL-15 even in the maximum available injection dose (equivalent 10 μg IL-15). In addition, we tested serum biochemical parameters and hematological parameters of mice treated with different formulations. It was found that IL-15 treatment (single dose >7.5 μg for four times) caused abnormal serum concentrations of aspartate transaminase (AST), alanine transaminase (ALT), and alkaline phosphatase (ALP), in contrast to biomimetic nanovaccine-mediated multivalent IL-15 self-transpresentation, for which the maximum achievable dosage still elicited a basal level of the biomarker (Fig. 4c–e). Meanwhile, biochemical analysis of blood harvested from IL-15 treated mice exhibited a remarkable decrease in creatinine (CREA), implying the possible nephrotoxicity induced by IL-15 (Fig. 4f). IL-15 also manifested the risk of thrombocytopenia with an evident reduction in platelet and red blood cell counts, whereas biomimetic nanovaccine-mediated IL-15 self-transpresentation displayed no overt toxicity even under the maximum administrable dosage (Fig. 4g, h). In summary, the

spatiotemporally synchronous presentation of IL-15 and tumor antigen prevented non-specific systemic immune stimulation, thus avoiding systemic side effects.

## In vivo antitumor activity

We next investigated the antitumoral activity of biNV-IL-15 in different tumor models (4T1 (Fig. 5a), CT26 (Supplementary Fig. 8a), and B16F10-OVA (Fig. 6a)) to assess the adaptability and flexibility of biomimetic nanovaccine-mediated spatiotemporal integration of multivalent IL-15 self-transpresentation and tumor antigen presentation. Tumor-bearing mice were treated with PBS, DC vesicles, IL-15, biNV, IL-15+biNV, and biNV-IL-15 on days 7, 10, 13, and 16. To this end, 4T1 or CT26 tumor-derived HCP and ovalbumin (OVA) served as the tumor antigens that target 4T1, CT26, and B16F10-OVA, respectively. The tumor volume was monitored every 2 days until day 40. As shown in Figs. 5b, 6b, Supplementary Figs. 4, 8b, c, 12, neither PBS nor DC vesicle treatment exhibited an antitumoral effect, and IL-15 delayed tumor growth slightly. While biNV displayed improved tumor inhibitory efficacy and prolonged survival, indicating the potential therapeutic effects of biomimetic nanovaccine-mediated tumor antigen presentation[45]. Compared with IL-15+biNV, biNV-IL-15 manifested dramatically enhanced anticancer activity and prolonged survival for over 49 days, which is likely due to its targeted delivery and local retention in secondary lymphatic organ, and spatiotemporally synchronous delivery of IL-15 and tumor antigen through biNV-IL-15 for enhanced tumor-specific T cell activation. Specifically, 3 out of 6 mice in the 4T1 tumor model and 4 out of 6 mice in CT26 and B16F10-OVA tumor models with the treatment of biNV-IL-15 showed complete tumor regression. To further evaluate the systemic toxicity of biNV-IL-15, histological analysis was performed by conducting the hematoxylin and eosin (H&E) staining of major organs (Supplementary Figs. 6, 11), while no noticeable pathological change was observed in the biNV-IL-15 group, demonstrating the biocompatibility of biNV-IL-15. Taken together, these results proved the effective tumor ablation and good safety profile of biNV-IL-15. Furthermore, the 4T1 tumor model was also established to compare the therapeutic outcomes of heterodimeric IL-15 (IL-15:IL-15Rα) and the MIST platform (Supplementary Fig. 9a). As shown in Supplementary Fig. 9b, the biNV-IL-15 exhibited superior antitumor activity compared with IL-15:IL-15Rα and IL-15:IL-15Rα+biNV, leading to prolonged survival period of mice.

Encouraged by the remarkable tumor elimination of biNV-IL-15, we further analyzed the profile of tumor-infiltrating lymphocytes (TIL) after different treatments. As shown in Fig. 5c, Supplementary Fig. 8d, IL-15 slightly increased the number of tumor-infiltrating CD8$^+$ T cells compared with PBS and DC vesicles. While higher infiltration of CD8$^+$ T cells was found after biNV treatment. In comparison with biNV, IL-15+biNV boosted effector T cell generation, resulting in stronger antitumor activity. Notably, the biNV-IL-15 treated group showed the highest CD8$^+$ T cell percentage, demonstrating that biNV-IL-15 treatment effectively activates tumor-infiltrating T cells. Immunofluorescence staining also indicated the increased tumor-infiltrating macrophages and CD8$^+$ T cells in the biNV-IL-15 group (Supplementary Figs. 5, 10). Proinflammatory cytokine secretion including TNF-α and IFN-γ assists CD8$^+$ T cell-mediated cancer elimination. We then investigated the secretion of IFN-γ in TME through ELISA analysis and determined TNF-α production in TME through immunohistochemical staining. The biNV-IL-15 treatment induced much higher TNF-α and IFN-γ secretion than IL-15+biNV in 4T1 and CT26 tumor models (Fig. 5g, h, Supplementary Fig. 8h, i). Meanwhile, immunosuppressive regulatory T cells (Treg) were consistently reduced after biNV-IL-15 therapy (Fig. 5c, Supplementary Fig. 8d), which might be attributed to the decrease in de novo induction of peripheral Tregs in TME or diminished recruitment of Tregs to TME. M2-like tumor-associated macrophages (TAM, CD206$^{hi}$CD11b$^+$F4/80$^+$) have been demonstrated to elicit Treg accumulation in TME either through Treg recruitment by

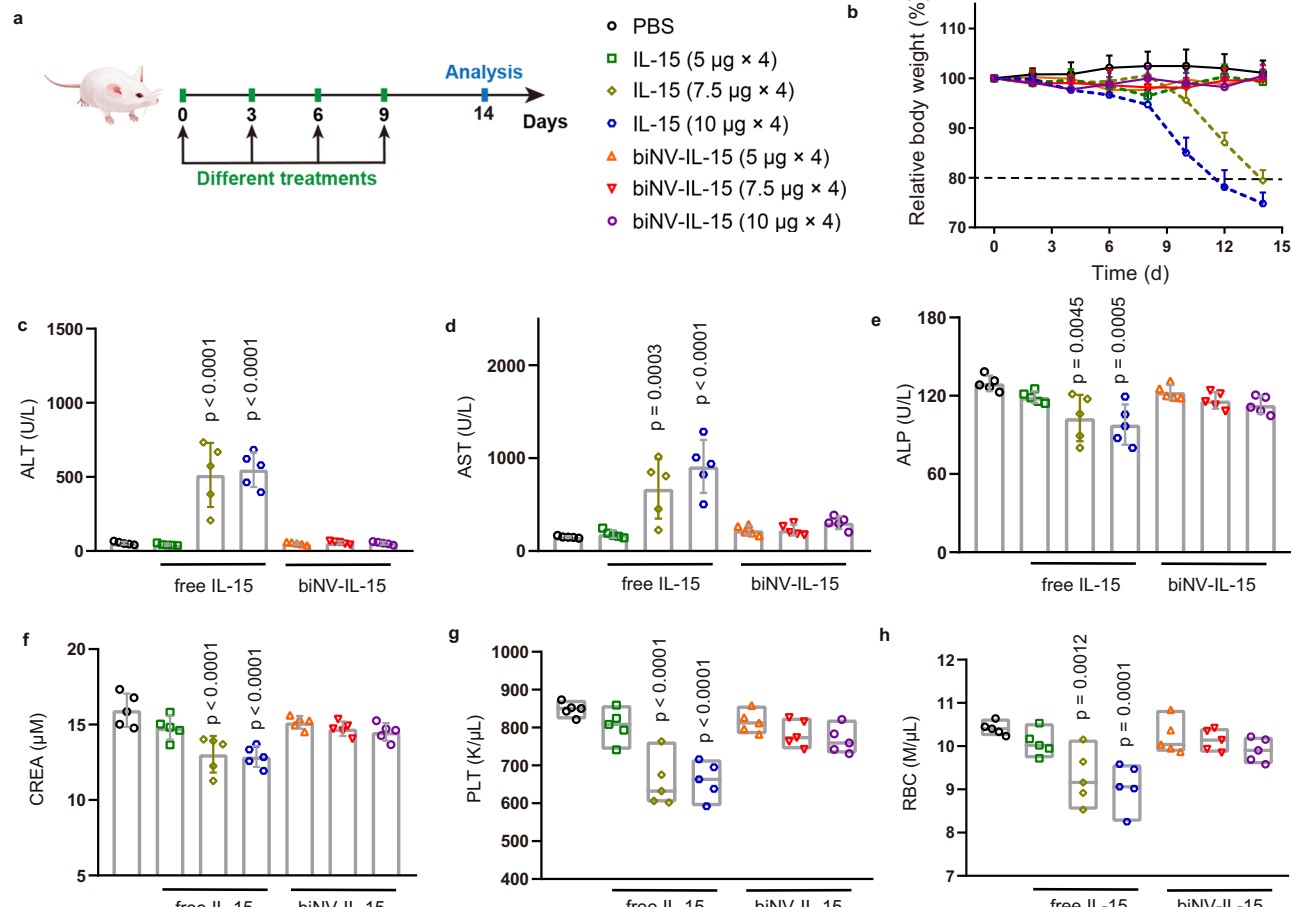

**Fig. 4 | The improved therapeutic window of IL-15 treatment. a** Normal BALB/c mice received intravenous administration of PBS, free IL-15, and biNV-IL-15 for multiple injections (days 0, 3, 6, and 9) at various doses. **b** Body weights were normalized to that on day 0, and the changes were recorded for 14 days following various treatments. The blood sample was taken for hematologic and biochemical analysis: **c** ALT, alanine aminotransferase (U/L), **d** AST, aspartate aminotransferase (U/L), **e** ALP, alkaline phosphatase (U/L), **f** CREA, creatinine (μM), **g** PLT, platelet count (K/μL), **h** RBC, red blood cell count (M/μL). The sample was harvested on day 14 or once the mice were euthanized because of the toxicity. **g**, **h** The box plots define the minima, maxima, center, and bounds of box. Data represent the mean ± s.d. (*n* = 5/group) and were compared with the PBS group for statistical analysis. The *p* values in panels **c**, **f**, **g** are <0.0001. And the *p* values of IL-15 (7.5 μg × 4) to PBS and IL-15 (10 μg × 4) to PBS in panel **d** are 0.0003 and <0.0001, respectively. Statistical significance was calculated through one-way ANOVA using a Tukey post-hoc test. Source data underlying panels **b**–**h** are provided as a Source Data file.

chemotaxis or peripheral Treg stimulation from naive CD4⁺ T cells. Flow cytometry analysis indicated that biNV-IL-15 induced phenotypic polarization of M2-like TAMs to M1-like TAMs (CD80^hiCD11b⁺F4/80⁺) (Fig. 5c, Supplementary Fig. 8d). Considering that type-1 cytokines including TNF-α and IFN-γ could mediate M1 transformation of TAMs, biNV-IL-15 may suppress the Treg accumulation in TME through converting TAM polarization to M1-type. To further confirm the immune priming, we investigated the role of biNV-IL-15 in T cell responses in draining lymph nodes (dLN). As shown in Fig. 5e and Supplementary Fig. 8f, the improved CD8⁺ T cell infiltration and reduced Tregs in dLNs indicated that biNV-IL-15 could enter LNs and effectively activate T cells through spatiotemporal integration of multivalent IL-15 self-transpresentation and tumor antigen presentation. Furthermore, we detected the proportion of CD8⁺ effector memory T cells (T_EM, CD3⁺CD8⁺CD62L^lowCD44^hi) in the spleen after various treatments through flow cytometry. As shown in Fig. 5d and Supplementary Fig. 8e, IL-15 alone elicited insufficient T_EM cell amplification in comparison with PBS and DC vesicles, and the T_EM cell expression in the spleen increased after the treatment of biNV. In stark contrast, biNV-IL-15 treatment yielded significantly more T_EM cells compared with IL-15+biNV treatment, implying that efficient T_EM cell induction may contribute to the long-lasting protective memory against cancer[46]. In

the B16F10-OVA tumor model, we further assessed the percentage of SIINFEKL-specific CD8⁺ T cells in peripheral blood through flow cytometry analysis of tetramer⁺CD8⁺ T cells following various treatments[47]. As shown in Fig. 6c, d, Supplementary Fig. 13, biNV induced only ~8% antigen-specific CTLs after the fourth immunization. Following IL-15+biNV treatment, the proportion of antigen-specific CD8⁺ T cells raised to a higher level of ~13%. Notably, the biNV-IL-15 evoked the peak frequency of ~25% antigen-specific CD8⁺ T cells, demonstrating a potentiated systemic T cell response. The flow cytometric assays were also performed in 4T1 tumor model established to compare the efficacy of heterodimeric IL-15 and the MIST platform. As shown in Supplementary Fig. 9c, e, biNV-IL-15 effectively enhanced the proportion of CD8⁺ T cells and decreased the immunosuppressive Tregs in tumors and dLNs, displaying obvious advantages over IL-15:IL-15Rα and IL-15:IL-15Rα+biNV. Meanwhile, biNV-IL-15 significantly remodeled TAMs from M2 type to M1 type and upregulated T_EM level in the spleen in comparison with IL-15:IL-15Rα and IL-15:IL-15Rα+biNV (Supplementary Fig. 9c, d).

To further validate the anticancer immunological memory induced by biNV-IL-15, the survived mice in the biNV-IL-15 group on day 55 were rechallenged with 4T1, CT26, and B16F10-OVA cancer cells, respectively. Meanwhile, naive mice of the same age were implanted

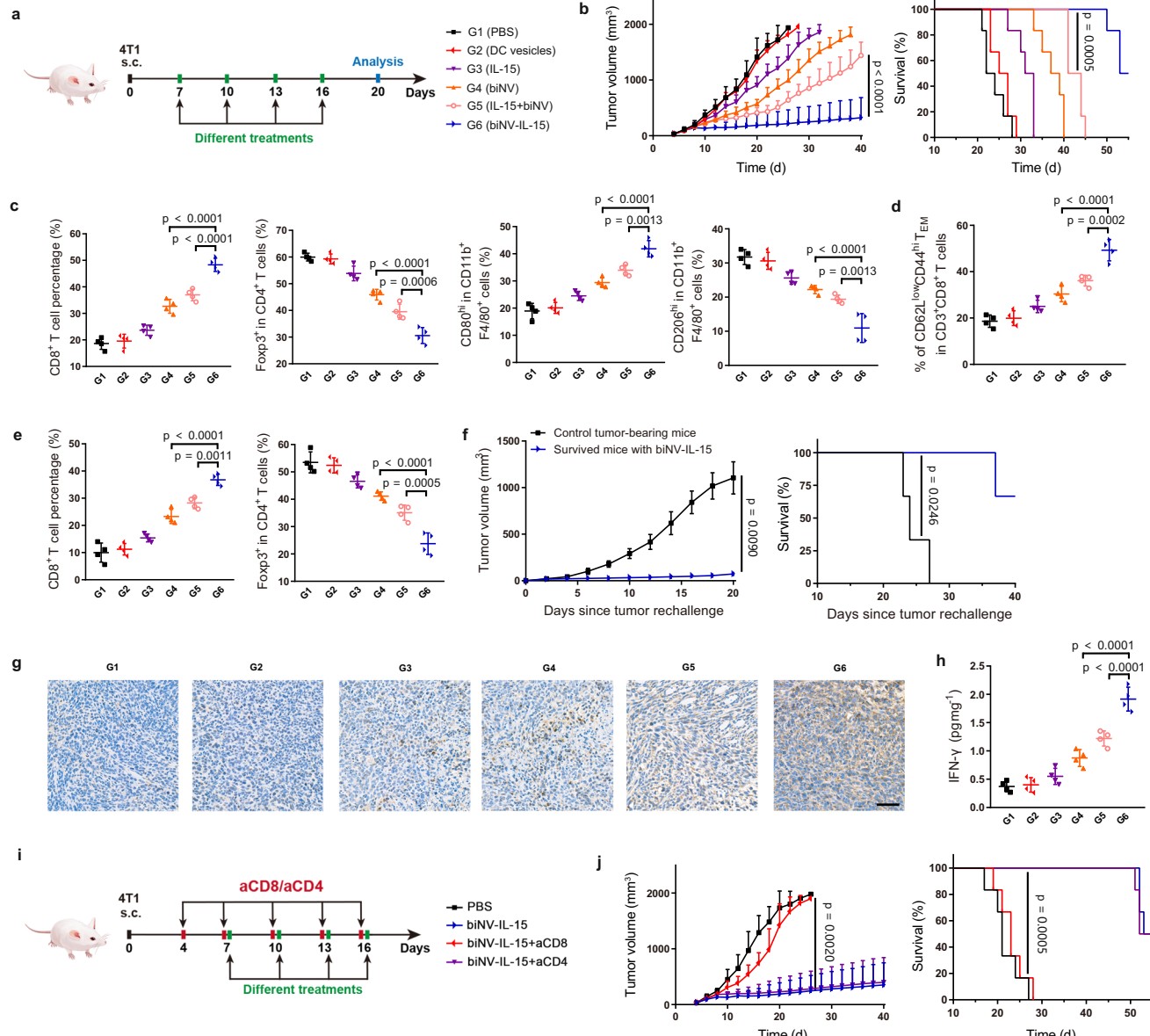

**Fig. 5 | In vivo antitumor activity of biomimetic nanovaccine-mediated multi-valent IL-15 self-transpresentation (MIST). a** 4T1 cancer cells were subcutaneously inoculated into BALB/c mice. On days 7, 10, 13, and 16, the PBS, DC vesicles, IL-15, biNV, IL-15+biNV, and biNV-IL-15 were intravenously administered into the mice. **b** Average tumor growth curve and survival curve following various treatments (n = 6/group). **c** Flow cytometric quantification of tumor-infiltrating CD8+ T cells, CD4+Foxp3+ Tregs, M1-type TAMs (CD80hiCD11b+F4/80+) and M2-type TAMs (CD206hiCD11b+F4/80+) in 4T1 tumor model (n = 4/group). **d** Flow cytometric quantification of CD3+CD8+CD62LlowCD44hi effector memory T cells (T_EM) in spleen (n = 4/group). **e** Flow cytometric quantification of CD8+ T cells and CD4+Foxp3+ Tregs in dLNs (n = 4/group). Survived mice in biNV-IL-15 group on day 55 were rechalleged with 4T1 cells. **f** The tumor growth curve and survival curve of 4T1

tumor rechallenged mice were recorded (n = 3/group). **g** Immunohistochemical staining for TNF-α of tumor sections following different therapies. Scale bar = 100 μm. **h** The secretion of IFN-γ in the 4T1 tumor was detected via ELISA assay (n = 4/group). **i** Experimental design for immune depletion in 4T1 tumor model. **j** Average tumor growth curve and survival curve (n = 6/group) of 4T1 tumor-bearing mice after biNV-IL-15 administration along with lymphocyte depletion. Data represent the mean ± s.d. The p values of G4 to G6 in panels **c, d, e, h** are <0.0001. And the p values of G5 to G6 in panels **c, d, e, h** are <0.0001, 0.0006, 0.0013, 0.0013, 0.0002, 0.0011, 0.0005, and <0.0001, respectively. Statistical significance was calculated through two-tailed student's t-test (**b, f**), log-rank (Mantel-Cox) test (**b, f, j**), or one-way ANOVA using a Tukey post-hoc test (**c–e, h, j**). Source data underlying panels **b–f, h, j** are provided as a Source Data file.

with the corresponding cell lines as a control. Surprisingly, survived mice in the biNV-IL-15 group were resistant to cancer rechallenge, and the survival time was significantly prolonged (Fig. 5f, Supplementary Figs. 8g, 14a, b). These results confirmed that biomimetic nanovaccine-mediated multivalent IL-15 self-transpresentation could lead to durable antitumor immunity.

We then investigated whether immune cell depletion affects the therapeutic effects of biNV-IL-15 (Fig. 5i, Supplementary Figs. 8j, 15a). The results showed that CD8+ T cell depletion almost completely

abolished tumor inhibition of biNV-IL-15. Instead, CD4+ T cell depletion did not change the treatment efficacy of biNV-IL-15, implying the central role of CD8+ T cells in biomimetic nanovaccine-mediated multivalent IL-15 self-transpresentation (MIST) (Fig. 5j, Supplementary Figs. 7, 8k, l, 15b, c).

To verify the antigen-specific therapeutic property of biNV-IL-15, 4T1 tumor-derived HCP and OVA served as the tumor antigens of 4T1 and B16F10-OVA. PBS, biNV-IL-15/HCP, or biNV-IL-15/OVA were intravenously administered into the 4T1 tumor- or B16F10-OVA tumor-

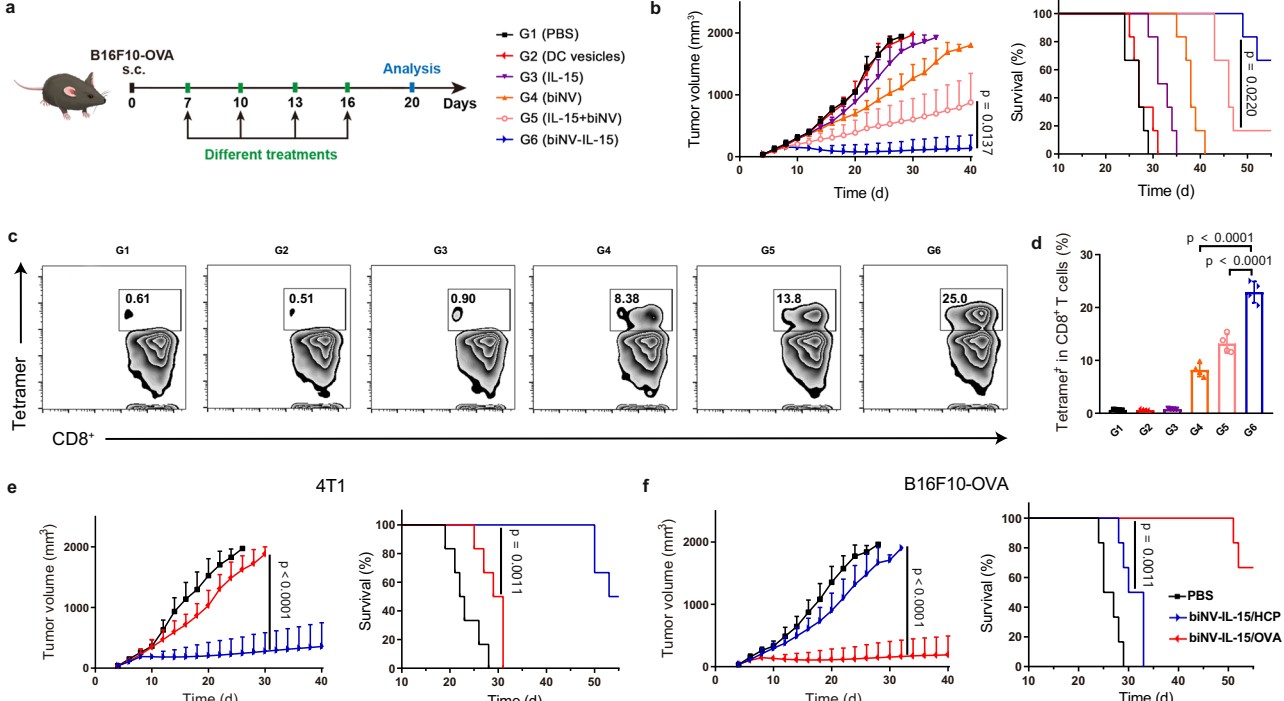

**Fig. 6 | In vivo antitumor activity of biomimetic nanovaccine-mediated multivalent IL-15 self-transpresentation (MIST). a** B16F10-OVA cancer cells were subcutaneously inoculated into C57BL/6j mice. On days 7, 10, 13, and 16, the PBS, DC vesicles, IL-15, biNV, IL-15+biNV, and biNV-IL-15 were intravenously administered into the mice. OVA served as the tumor antigen of B16F10-OVA. **b** Average tumor growth curve and survival curve following various treatments ($n = 6$/group). **c** Flow cytometric examination images and **d** relative quantification of SIINFEKL-specific CD8$^+$ T cells in peripheral blood of B16F10-OVA tumor-bearing mice after various treatments, which was monitored through flow cytometry analysis of tetramer$^+$CD8$^+$ T cells ($n = 4$/group). Average tumor growth curve and survival curve ($n = 6$/group) of 4T1 (**e**) and B16F10-OVA (**f**) tumor-bearing mice suggested the in vivo antigen-specific cancer inhibition of biNV-IL-15. Data represent the mean ± s.d. The $p$ values in panel **d** are <0.0001. And the $p$ values of biNV-IL-15/HCP to biNV-IL-15/OVA in panel **f** are <0.0001 and 0.0011, respectively. Statistical significance was calculated through two-tailed student's $t$-test (**e, f**), log-rank (Mantel-Cox) test (**b, e, f**), or one-way ANOVA using a Tukey post-hoc test (**b, d**). Source data underlying panels **b, d, e, f** are provided as a Source Data file.

bearing mice on days 7, 10, 13, and 16. The results showed that biNV-IL-15/HCP remarkably delayed the growth of 4T1 cancer, whereas the biNV-IL-15/OVA did not affect cancer suppression (Fig. 6e, Supplementary Fig. 16a). Similarly, the growth of B16F10-OVA cancer was restrained by biNV-IL-15/OVA treatment instead of biNV-IL-15/HCP treatment (Fig. 6f, Supplementary Fig. 16b). Collectively, these findings indicated that biNV-IL-15 could produce CD3$^+$CD8$^+$ CTLs in an antigen-specific manner in vivo via the spatiotemporally synchronous provision of IL-15 and tumor antigen to CD8$^+$ T cells.

## The suppression of tumor recurrence

Despite the advancement of surgical techniques, tumor recurrence remains a clinical challenge due to the residual and circulating tumor cells[48]. Considering the ability of biNV-IL-15 to stimulate antitumor immunity, we constructed the cancer resection model to validate the anti-recurrence effects of biomimetic nanovaccine-mediated IL-15 self-transpresentation (Fig. 7a). Tumor was resected with ~1% left to simulate residual microtumor after surgery. The tumor recurrence could be recorded by bioluminescence signal from 4T1-luc cells, and recurrent tumor volume was measured using a caliper[49]. As shown in Fig. 7b-e, IL-15 alone failed to suppress tumor recurrence with unsatisfactory survival. By contrast, the bioluminescence intensities of mice in the biNV group decreased to some extent. Moreover, IL-15+biNV further delayed tumor recurrence in comparison with biNV. Strikingly, once biNV-IL-15 was administered after surgery, 3 out of 6 mice displayed no visible tumor at the primary inoculation site, resulting in a high survival rate for long-period investigation. These results suggested that biNV-IL-15 could effectively suppress post-surgical tumor recurrence, which may be attributed to the exclusive and valid antitumor immunity

induced by spatiotemporally synchronous delivery of IL-15 and tumor antigen. Furthermore, the tumor resection model was also established to compare the anti-recurrence effects of heterodimeric IL-15 and the MIST platform (Supplementary Fig. 17a). As shown in Supplementary Fig. 17b−d, the biNV-IL-15 displayed the excellent capability of tumor recurrence inhibition in contrast to IL-15:IL-15Rα and IL-15:IL-15Rα +biNV, resulting in superior survival outcomes.

We next harvested the residual tumors following different treatments to analyze TILs with flow cytometry. Compared with other controls, the biNV-IL-15 treatment further upregulated CD8$^+$ T cell percentage, while CD4$^+$Foxp3$^+$ Treg percentage was reduced (Fig. 7g−j, Supplementary Fig. 18). Moreover, Fig. 7k−n, Supplementary Figs. 19, 20 displayed that the percentage of M2-like TAMs was decreased, and that of M1-like TAMs was elevated following biNV-IL-15 therapy. The phenotypic polarization of M2-like TAMs to M1-like TAMs could block the TAM-mediated tumor blood vessel growth and lymph vessel formation, and finally suppress tumor metastasis and recurrence. Consistently, immunofluorescence staining visualized the recruitment of macrophages and CD8$^+$ T cells in residual tumor tissue after biNV-IL-15 treatment (Fig. 7f). We also detected the levels of IFN-γ and TNF-α in TME via ELISA analysis and immunohistochemical staining, respectively. The increased cytokine levels in TME suggested the efficient facilitation of adaptive immune responses following biNV-IL-15 treatment (Supplementary Fig. 21a, b). Furthermore, we monitored the differences in T$_{EM}$ cell percentage after various treatments. As shown in Fig. 7o, p, Supplementary Fig. 22, biNV-IL-15 therapy significantly enhanced the T$_{EM}$ cell level in the secondary lymphoid tissue (spleen), confirming that the biomimetic nanovaccine can evoke immune memory. The flow cytometric assays were also performed in tumor

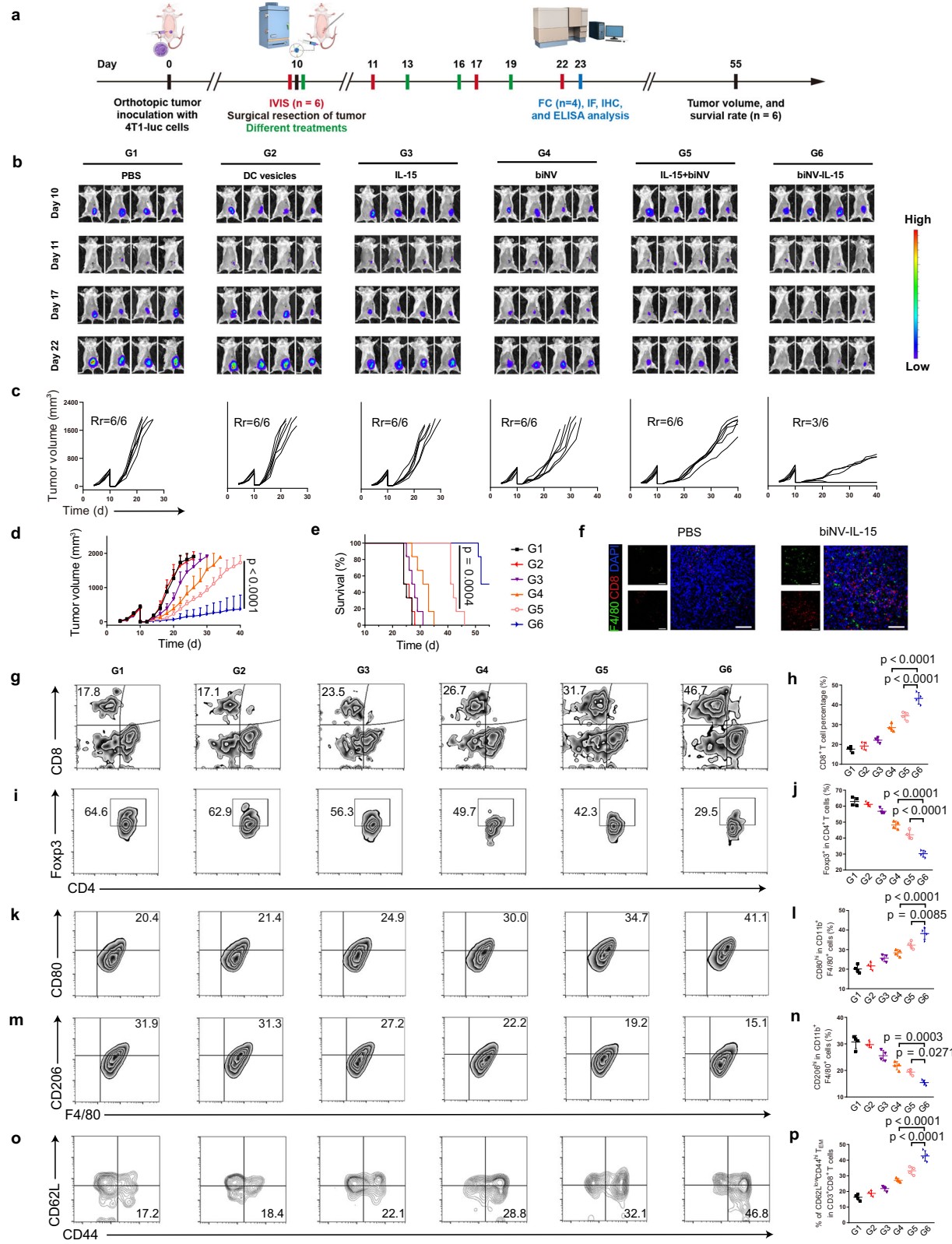

resection model established to compare the efficacy of heterodimeric IL-15 and the MIST platform. As shown in Supplementary Fig. 17e−h, biNV-IL-15 obviously improved the percentage of tumor-infiltrating CD8⁺ T cells and downregulated the level of immunosuppressive Tregs in tumor, implying remarkable advantages over IL-15:IL-15Rα and IL-15:IL-15Rα+biNV. Furthermore, biNV-IL-15 effectively polarized M2-like TAMs to M1-like TAMs and increased splenic $T_{EM}$ proportion in

contrast to IL-15:IL-15Rα and IL-15:IL-15Rα+biNV (Supplementary Fig. 17i−n).

## In vivo anti-metastasis performance

It has been reported that circulating cancer cells could invade various organs, especially the lung, leading to cancer metastasis[50]. In our study, we intravenously injected 4T1-luc cancer cells into the

**Fig. 7 | The suppression of tumor recurrence. a** Schematic depicting biNV-IL-15 treatment in 4T1-luc orthotopic cancer model with incomplete cancer resection (IVIS in vivo imaging system, FC flow cytometry analysis, IF immunofluorescence, IHC immunohistochemistry, ELIS: enzyme-linked immunosorbent assay). **b** In vivo bioluminescence imaging for 4T1-luc tumor following the removal of the primary tumor. Every group showed four representative mice. Images on day 10 were shown before surgery. **c** Individual and **d** average tumor growth curve and **e** survival curve ($n = 6$/group) in tumor resection model receiving various treatments, Rr: Recurrence rate. **f** Immunofluorescence images of tumors displaying F4/80$^+$ macrophage and CD8$^+$ T cell infiltration for PBS and biNV-IL-15 groups. Scale bar = 50 μm. Experiment was repeated three times independently with similar results. **g** Flow cytometric examination images and **h** relative quantification of CD8$^+$ T cell in tumor ($n = 4$/group). **i** Flow cytometric measurement images and **j** relative quantification of CD4$^+$Foxp3$^+$ Tregs in tumor ($n = 4$/group). **k** Flow cytometric assessment images and **l** relative quantification of M1-type TAMs (CD80$^{hi}$CD11b$^+$F4/80$^+$) in tumor ($n = 4$/group). **m** Flow cytometric analysis images and **n** relative quantification of M2-type TAM (CD206$^{hi}$CD11b$^+$F4/80$^+$) in tumor ($n = 4$/group). **o** Flow cytometric evaluation images and **p** relative quantification of CD3$^+$CD8$^+$CD62L$^{low}$CD44$^{hi}$ T$_{EM}$ in the spleen ($n = 4$/group). Data represent the mean ± s.d. The $p$ values in panels **d**, **h**, **j**, **p** are <0.0001. And the $p$ values of G4 to G6 and G5 to G6 in panel **l** are <0.0001 and 0.0085, respectively. Statistical significance was calculated through two-tailed student's $t$-test (**d**), log-rank (Mantel-Cox) test (**e**), or one-way ANOVA using a Tukey post-hoc test (**h**, **j**, **l**, **n**, **p**). Source data underlying panels **c**–**e**, **h**, **j**, **l**, **n**, **p** are provided as a Source Data file.

orthotopic tumor-bearing mice to simulate malignant tumor invasion and hematogenous metastasis (Fig. 8a)[51]. The additional tail vein injection of tumor cells into tumor-bearing mice to simulate hematogenous metastasis has been widely applied as an artificial whole-body spreading tumor model[52–56]. Compared with spontaneous lung metastasis, the whole-body metastasis model was more aggressive and challenging, which was suitable for specialized anti-metastasis evaluation. The tumor growth curve and survival curve were recorded and lungs were excised to ex vivo analyze the extent of metastasis with bioluminescence imaging. As shown in Fig. 8b–e, Supplementary Fig. 23, the PBS and DC vesicle treated mice displayed severe metastatic foci in the lung. The bioluminescence intensities and primary tumor growth of mice treated with IL-15 exhibited a slight reduction. In contrast, the biNV allowed an improved metastasis suppression effect. Compared with biNV, IL-15+biNV further relieved lung metastasis and tumor progression. It is noteworthy that biNV-IL-15 could effectively eliminate the metastasis signal in the lung and remarkably inhibit primary tumors. The metastasis sites could also be detected via Bouin's fluid staining (Fig. 8f, g). Consistently, the mice treated with biNV-IL-15 showed negligible signs of metastasis in Bouin's fluid staining, implying effective inhibition of cancer metastasis to the lung, and the degree of metastasis in the lung varied among other groups. Additionally, the lung and liver invasion of cancer cells was evaluated with H&E staining. The dashed outlines indicated metastases and the ratio of the metastatic area was calculated. As shown in Fig. 8h–k, cancer cells seriously invaded both lung and liver in PBS and DC vesicle groups, and IL-15 indicated a weak metastasis-inhibitory ability. By contrast, biNV produced an evident reduction of metastatic nodules. In comparison with IL-15+biNV, biNV-IL-15 suppressed the development of cancer metastasis much more effectively, achieving synergistic therapeutic outcomes of spatiotemporally synchronous provision of IL-15 and tumor antigen to CD8$^+$ T cells. Moreover, the hematogenous metastasis model was also established to compare the antimetastatic ability of heterodimeric IL-15 and the MIST platform (Supplementary Fig. 24a). As shown in Supplementary Fig. 24b–k, the biNV-IL-15 treatment not only inhibited the tumor growth but also prevented lung and liver metastasis of the tumor in contrast to IL-15:IL-15Rα and IL-15:IL-15Rα+biNV, causing an extended survival time. To clarify the anti-metastasis mechanism of biNV-IL-15, we investigated the systemic antitumor immune responses after various treatments. As assessed by flow cytometry, the biNV-IL-15 dramatically augmented CTLs and reduced Tregs (Fig. 8l–o). In general, the biomimetic nanovaccine-mediated multivalent IL-15 self-transpresentation (MIST) could trigger T cell-mediated cancer cell destruction in blood circulation to suppress systemic development. The flow cytometric assays were also performed in hematogenous metastasis model established to compare the metastasis-inhibitory ability of heterodimeric IL-15 and the MIST platform. As shown in Supplementary Fig. 24l–o, biNV-IL-15 significantly elevated the level of CD8$^+$ T cells and decreased the proportion of Tregs in

blood, demonstrating distinctive advantages over IL-15:IL-15Rα and IL-15:IL-15Rα+biNV.

## Discussion

In summary, we elaborately constructed a cytokine delivery platform capable of preventing systemic side effects and activating antitumor immunity. Leveraging on the outstanding safety profile and well-established fabricating processes, the biomimetic nanovaccine exhibits dramatic promise to improve the targeted delivery of IL-15 to lymph nodes. Mechanistically, we have illustrated its ability to induce IL-15 self-transpresentation in a multivalent manner and activate robust tumor-specific CTL responses. Conventional cytokine therapy often elicits dose-limiting toxicity attributed to the systemic activation of nontumor-specific T cells. This approach can revolutionize traditional cytokine therapy by inducing safer and more efficient CTL immune responses against tumors by spatiotemporal integration of multivalent IL-15 self-transpresentation (MIST) and tumor antigen presentation. This strategy augments the tumor-specific T-cell responses, exerts significantly enhanced therapeutic effects, facilitates tumor clearance, and raises persistent immunological memory. Given the tolerance of cytomembrane vesicles to the insertion of natural proteins, this strategy offers a generalizable approach for cytokine delivery. Despite the somewhat complicated manufacturing procedure of cytomembrane vesicle formulation, the production and quality control of cytomembrane vesicles is not the main obstacle since we have set up criteria for the scale-up preparation of cytomembrane vesicles with proper quality control. As a result, such a biomimetic nanovaccine platform technology provides a modality for the practical clinical application of cytokine therapy.

## Methods

### Ethical statement

Our research complies with all relevant ethical regulations. All the animal protocols were performed in line with the Guidelines for the Care and Use of Laboratory Animals and approved by the Institutional Animal Ethical Care Committee (IAEC) of Shenyang Pharmaceutical University.

### Materials

The IL-15 and IL-15Rα encoded rAD was constructed by Shanghai Genechem Co., Ltd, China. Recombinant mouse IL-15 was provided by Abcam Inc., USA. Pierce$^{TM}$ Protein A/G Agarose and ATP solution were obtained from ThermoFisher Scientific, USA. Red blood cell lysis buffer was supplied by Beijing Solarbio Science & Technology Co., Ltd, China. Mouse IL-15 ELISA kit was purchased from Neobioscience Technology Co, Ltd, China. Mouse IFN-γ ELISA kit was obtained from Shanghai Jianglai Industrial Limited by Share Ltd, China. OVA was supplied by InvivoGen, USA. Sulfo-Cy7 NHS ester was obtained from Lumiprobe Corporation, USA. LPS was bought from Sigma-Aldrich, USA. Cell culture flasks, plates, and glass bottom culture dishes were purchased from Wuxi NEST Biotechnology Co., Ltd, China. Scientz-IID

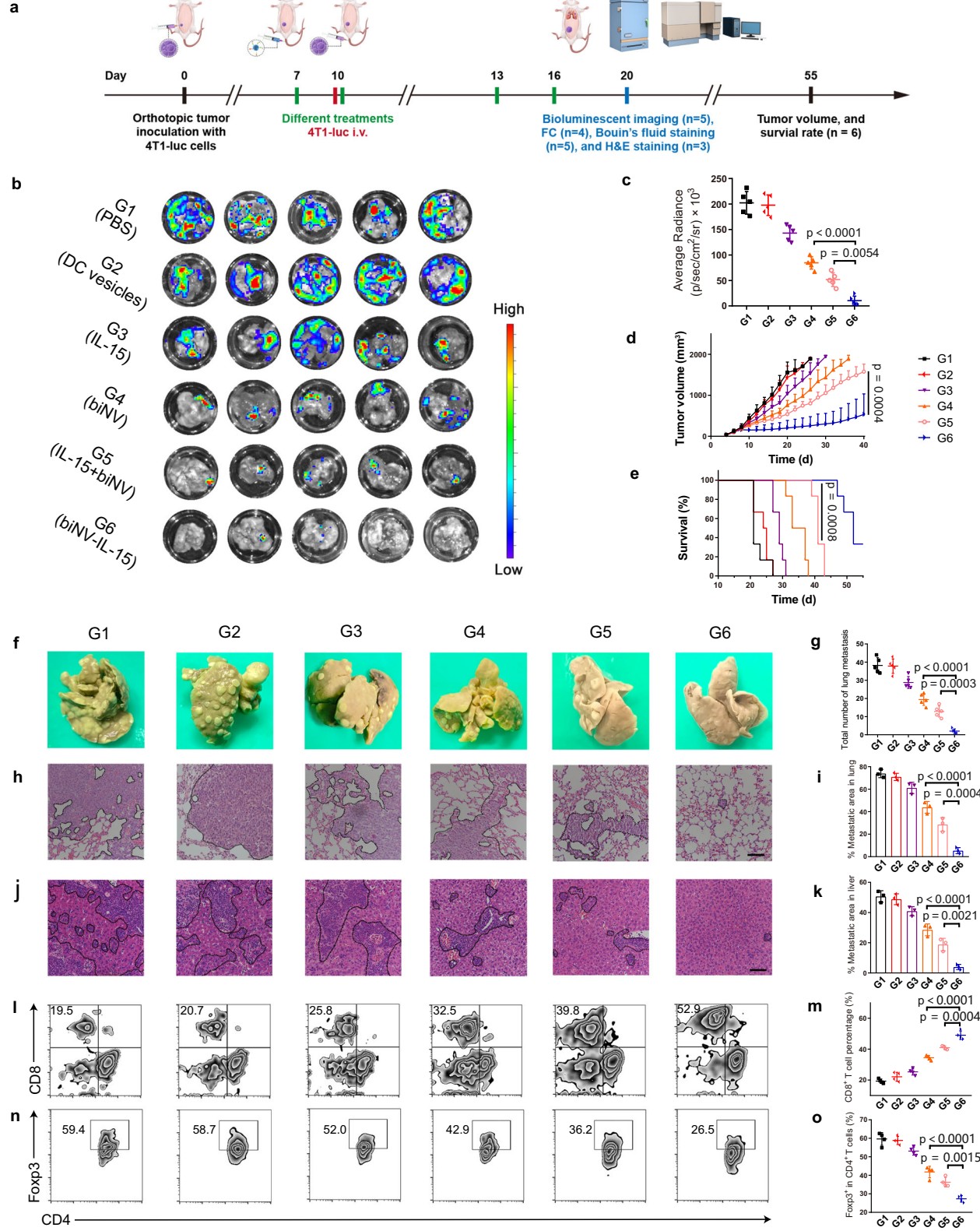

**Nature Communications** | (2023)14:6748

Ultrasonic Homogenizer and Gene Electroporator Scientz-2C were provided by Ningbo Scientz Biotechnology Co., Ltd, China. All used reagents and solvents were of analytical standard grade unless stated otherwise.

**Extraction and characterization of HCP**

4T1 or CT26 cancer tissue was dissociated through mechanical dissection and utilized to prepare a single-cell suspension with the 70 μm cell strainer. Red blood cell lysis buffer was added to discard red blood cells. The prepared cancer cell was then homogenized on ice for 1 h in lysis buffer including Protease Inhibitor Cocktail Set I. Following centrifugation at 4 °C for 30 min, the supernatant was immunoprecipitated using anti-HSP70 antibody (Santa Cruz, sc-24) and incubated with protein A/G-Sepharose beads for 12 h. The lysis buffer was then used to wash the protein/antibody/beads complex. The complex was incubated with ATP at 25 °C for 30 min. Afterwards, the supernatant

**Fig. 8 | In vivo anti-metastasis performance. a** 4T1-luc cancer cells were intravenously injected into the tumor-bearing mice on day 10 to simulate the hematogenous metastasis model (FC: flow cytometry analysis; H&E staining: hematoxylin and eosin staining). PBS, DC vesicles, IL-15, biNV, IL-15+biNV, and biNV-IL-15 were individually administrated on days 7, 10, 13, and 16. **b** Ex vivo bioluminescent imaging and **c** average bioluminescent radiance for isolated lungs were studied after various treatments on day 20 ($n = 5$/group). **d** Average tumor growth curve and **e** survival curve ($n = 6$/group) in hematogenous metastasis model receiving various treatments. **f** Photographs and **g** quantification of lung metastases ($n = 5$/group) for Bouin's fluid staining of lungs. H&E staining of the lung (**h**) and liver (**j**) slices after different treatments. Scale bar = 100 μm. Quantification for metastasis area percentages of the lung (**i**) and liver (**k**) slices ($n = 3$/group). **l** Flow cytometric analysis images and **m** relative quantification of CD8$^+$ T cells in the blood ($n = 4$/group). **n** Flow cytometric examination images and **o** relative quantification of CD4$^+$Foxp3$^+$ Tregs in the blood ($n = 4$/group). Data represent the mean ± s.d. The $p$ values of G4 to G6 in panels **c**, **g**, **i**, **k**, **m**, **o** are <0.0001. And the $p$ values of G5 to G6 in panels **c**, **g**, **i**, **k**, **m**, **o** are 0.0054, 0.0003, 0.0004, 0.0021, 0.0004, and 0.0015, respectively. Statistical significance was calculated through one-way ANOVA using a Tukey post-hoc test (**c**, **g**, **i**, **k**, **m**, **o**), two-tailed student's $t$-test (**d**) or log-rank (Mantel-Cox) test (**e**). Source data underlying panels **c**–**e**, **g**, **i**, **k**, **m**, **o** are provided as a Source Data file.

was gathered and quantitative measurement of protein was performed with Bradford assay. The protein was fractionated by SDS-PAGE electrophoresis and the gel was stained with Coomassie blue. The diameter of isolated HCP was measured using dynamic light scattering (DLS, Malvern, U.K.) with Zetasizer Software 7.01.

## Generation of biNV-IL-15

Bone marrow-derived dendritic cells (BMDCs) were isolated from BALB/c or C57BL/6j bone marrow according to the protocol reported previously. Briefly, we flushed and dispersed bone marrow cells from mice femurs and lysed red blood cells. Cells were cultured in RPMI-1640 (Gibco) supplemented with 100 μg mL$^{-1}$ streptomycin (Invitrogen), 100 U mL$^{-1}$ penicillin (Invitrogen), 10% fetal bovine serum (FBS, Gibco), 5 ng mL$^{-1}$ murine IL-4 (ThermoFisher) as well as 10 ng mL$^{-1}$ murine GM-CSF (ThermoFisher). Half changed the medium every 2 d. The DCs ($1 \times 10^6$ cells/well) were collected and plated in 12-well plate on day 5, and infected with a combination of IL-15 and IL-15Rα encoded recombinant adenovirus (IL-15/IL-15Rα rAD) at a multiplicity of infection (MOI) of 200 for 24 h in complete RPMI-1640 without IL-4 or GM-CSF at 37 °C with 5% CO$_2$. Then, continued to culture in complete RPMI-1640 replenished with cytokines (5 ng mL$^{-1}$ IL-4 and 10 ng mL$^{-1}$ GM-CSF) under the conditions of 5% CO$_2$ and 37 °C. On day 7, immature DCs were next pulsed with endogenous HCP containing 1 μg mL$^{-1}$ LPS and further cultured in RPMI-1640 for 24 h to acquire membrane-bound IL-15 anchored DCs with tumor antigen presentation (DC-IL-15/HCP). Following our previous study, we gathered and washed the DC-IL-15/HCP twice in cold PBS containing protease inhibitor for removing cell debris. The DC-IL-15/HCP were resuspended in PBS and sonicated in the sterile EP tube (22 W, 60 s) on ice. Next, the cytomembrane vesicles were collected by multistep density gradient ultracentrifugation, and suspended in PBS. We then introduced the isolated biNV-IL-15 to the Mini-Extruder (Avanti Polar Lipids) with 200 nm pore-sized membrane filter in PBS to acquire uniform cytomembrane vesicles. Meanwhile, DC vesicles were fabricated from BMDCs without any treatment.

## Characterization of biNV-IL-15

The western blotting examination was used to investigate the presence of major functional membrane proteins on biNV-IL-15. In brief, the protein molecules were separated using SDS-PAGE electrophoresis, and transferred protein to PVDF membrane. Next, blocked membrane for 1 h with 5% skim milk and incubated with primary antibodies of anti-Na/K ATPase (Abcam, ab254025, dilution: 1:1000), anti-B7-1 (Bioss, bs-1479R, dilution: 1:1000), anti-B7-2 (ABclonal, A19026, dilution: 1:1000), anti-ICAM-1 (Invitrogen, MA5407, dilution: 1:250), anti-CCR7 (Invitrogen, MA1-163, dilution: 1:500), anti-IL-15 (Abcam, ab273625, dilution: 1:1000) and anti-IL-15Rα (Invitrogen, PA5-114215, dilution: 1:500) at 4 °C, respectively. After incubation with secondary antibody for 1 h at room temperature, Western ECL Substrate was added to monitor horseradish peroxidase (HRP) conjugates on immunoblots. Uncropped and unprocessed full scan images of all western blots can be found in the Source Data file. The nanoparticle tracking analysis (NTA) was carried out with ZetaView (Particle Metrix, Germany). The morphology

of DC vesicles was detected with transmission electron microscopy (TEM, JEM-2100s) through uranyl acetate negative staining. Meanwhile, immunolabeling with antibody against MHC-I was also performed for electron microscopy. Specifically, the sample was incubated with anti-MHC-I antibody (Abcam, ab281901, dilution: 1:100) at 25 °C for 2 h, and washed in BSA solution 3 times. Following incubation with 10 nm gold conjugated goat anti-rabbit IgG (Bioss, bs-0295G-gold, dilution: 1:100) at 25 °C for 1 h, three washes in BSA were achieved. The sample was fixed with uranyl acetate and observed with TEM (JEM-2100s). To prove the co-expression of MHC-I and IL-15 on biNV-IL-15, GFP-tagged IL-15/IL-15Rα rAD was used to transduce BMDCs for DC vesicle isolation. DC vesicles were then incubated with anti-MHC-I antibody and further labeled with Alexa Fluor 594 conjugated secondary antibody (Invitrogen, A-11012, dilution: 1:200). The fluorescence was detected by confocal laser scanning microscopy (CLSM, C2SI, Nikon, Japan) using NIS 4.13. Moreover, the amount of IL-15 in DC vesicle lysates was detected with an enzyme-linked immunosorbent assay (ELISA) kit following the dissociation of IL-15/IL-15Rα complex by 0.01 % SDS and boiling. To determine the in vitro stability, biNV-IL-15 was resuspended at a concentration of 2 mg mL$^{-1}$ in PBS, and kept at 4 °C or 37 °C. The diameter was detected through DLS at predetermined time points ($n = 5$). Meanwhile, the particle size of biNV-IL-15 before and after cryopreservation at −80 °C was also measured for 60 d.

## Preparation of IL-15:IL-15Rα

IL-15:IL-15Rα, a heterodimer cytokine composed of IL-15 and IL-15Rα, was produced and purified from IL-15 and IL-15Rα transfected HEK293 cells. The cells were cultured in a serum-free medium (Lonza).

## Cell culture

4T1 cells (Serial: TCM32) and CT26 cells (Serial: TCM37) were supplied by the Chinese Academy of Sciences. 4T1 luciferase labeled cells (4T1-luc, Serial: LZQ0016) were obtained from Shanghai Zhongqiao Xinzhou Biotechnology Co., Ltd. B16F10-OVA cells, a variant of the B16-F10 murine melanoma cell line that expresses OVA, were kindly gifted from Professor Cong Luo, Shenyang Pharmaceutical University. Cell line validation with short tandem repeat (STR) markers was conducted via Genetic Testing Biotechnology Corporation (Suzhou, China). In detail, eighteen STR loci were amplified using multiplex PCR. One additional marker (Human TH01) was used to screen for the presence of human species. The cell line sample was processed with ABI Prism 3130 XL Genetic Analyzer. Data were analyzed by Gene Mapper ID 3.2 software (Applied Biosystems). Appropriate positive and negative controls were run and confirmed for sample. For CD8$^+$ T lymphocyte extraction, the spleen of BALB/c mice was digested with DNase I (SIGMA-ALDRICH) and Liberase (Roche Diagnostics) for producing single-cell suspension, followed by staining using R-phycoerythrin (PE)-conjugated anti-CD8 antibody and applying to anti-PE microbeads (Miltenyi Biotec) for extraction. The CD8$^+$ T lymphocytes as well as 4T1, 4T1-luc, and CT26 cancer cell lines were cultured in RPMI-1640 (Gibco) and B16F10-OVA cell line was cultured in high-glucose DMEM (Gibco) enriched with 100 μg mL$^{-1}$ streptomycin (Invitrogen), 100 U mL$^{-1}$ penicillin

(Invitrogen) and 10% FBS (Gibco) under the conditions of 5% $CO_2$ and 37 °C. Throughout the research, cells were utilized as received and examined negative for rodent pathogens and mycoplasma contamination.

## The direct activation of naive CD8$^+$ T cells

HCP used to pulse DCs was isolated from 4T1 and CT26 tumor tissue, respectively. The biNV-IL-15 was then extracted from DC-IL-15/HCP as described above. We investigated the capability of biNV-IL-15 for multivalent IL-15 self-transpresentation and T cell response activation. The $2 \times 10^5$ splenic CD8$^+$ T lymphocytes were treated with DC vesicles, IL-15, biNV, IL-15+biNV, and biNV-IL-15 (5 µg mL$^{-1}$ DC vesicles or 250 ng mL$^{-1}$ IL-15). Following 7 d incubation, TNF-α or IFN-γ spot-forming cells were analyzed by enzyme-linked immune absorbent spot (ELISPOT) assay. Specifically, transferred cells to ELISPOT plate (Merck), which was coated with anti-TNF-α or anti-IFN-γ (Mabtech). Following incubation for 24 h, washed away cells and incubated plate with biotinylated anti-TNF-α or anti-IFN-γ antibody (Mabtech) at 25 °C for 2 h. Afterwards, washed plate using PBS as well as incubated with streptavidin-alkaline phosphatase (Mabtech) at 25 °C for 1.5 h. After washing using PBS, BCIP/NBT substrate solution (CALBIOCHEM) was added to the plate at 25 °C for about 1 h. Stopped color development via washing using tap water. Following drying, counted spots through an ELISPOT reader with AID ELISPOT software (version 5.0). Subsequently, Cytotoxicity Detection Kit (LDH) (Roche, USA) was utilized to assess the cytolysis efficiency against 4T1 and CT26 cancer cells. Activated T lymphocytes (effector cells) were cocultured with two types of cancer cells (target cells) at 37 °C for 4 h with effector-to-target cell (E: T) ratios of 5:1, 10:1, 20:1, and 40:1. The spontaneous release of lactate dehydrogenase (LDH) by cancer cells (low control) was evaluated via no T cell addition. The maximum release of LDH by cancer cells (high control) was detected via incubation of cancer cells with 1% Triton X-100. The absorption was measured at 490 nm using microplate reader (SkanIt 2.4.3.37) after incubation of supernatant with LDH Cytotoxicity Detection reagent (reference wavelength: 630 nm). Cytolysis efficiency (%) was determined using the equation below: cytotoxicity (%) = (T/cancer cells mix - T cells along - low control) / (high control - low control) × 100 %. To explore the T lymphocyte proliferation, CD8$^+$ T cell was stained with CFSE per the manufacturer's procedure. DC vesicles, IL-15, biNV, IL-15+biNV, and biNV-IL-15 (5 µg mL$^{-1}$ DC vesicles or 250 ng mL$^{-1}$ IL-15) were cocultured with CFSE-labeled T cells for 72 h, respectively. Flow cytometry (BD CellQuest Pro v5.2) was applied to detect the proliferation of T lymphocytes by CFSE labeling.

## Animal studies

The BALB/c and C57BL/6j mice (female, 6–8 weeks old) were supplied by the Animal Center of Shenyang Pharmaceutical University (Shenyang, Liaoning, China). Because one of the selected model was breast cancer, female animals were selected for experiments. Although we have used single-sex animals in our research, we think that the research results were not only applicable to single sex. Mice were bred and maintained in the specific pathogen-free (SPF) animal facility. Experimental/control animals were co-housed and spread across cages. The living environment of animals were maintained at a temperature of ~25 °C and a humidity of 50 ± 5% with a 12 h light/dark cycle, with free access to standard food and water. The humane endpoints included tumor burden exceeding 10% of normal body weight, animal weight loss exceeds 20% of normal animal weight, ulcer at tumor growth point, and sustained self-mutilation in animals. The humane end-point was approved by Certification and Accreditation Administration of the People's Republic of China (CNCA). Cervical dislocation under deep anesthesia was adopted for euthanasia. All experiments and procedures were conducted following the Guide for Care and Use of Laboratory Animals which was approved by the Institutional Animal Ethical Care Committee (IAEC) of Shenyang Pharmaceutical University.

## Pharmacokinetic and biodistribution study of Cy7-labeled DC vesicles

Female BALB/c mice aged 6–8 weeks were applied and acclimatized to the animal center. For labeling the IL-15, sulfo-Cy7 NHS ester was mixed with IL-15, and the mixture was incubated in the dark. After the labeling reaction, the unbinding dye was removed using a 7 kDa Zeba buffer exchange column (ThermoFisher) to elute the labeled IL-15 (IL-15-Cy7). To label cytomembrane vesicles with Cy7, sulfo-Cy7 NHS ester was added to biNV-IL-15. Exosome Spin Column (MW 3000) (ThermoFisher) was used to purify Cy7-labeled cytomembrane vesicles (biNV-IL-15-Cy7), followed by centrifugation at $3500 \times g$ for 16 min. For the pharmacokinetic study, the mice were intravenously injected with IL-15-Cy7 or biNV-IL-15-Cy7 ($n = 3$). The blood sample was harvested at timed intervals (2 min, 5 min, 15 min, 30 min, 1 h, 2 h, 4 h, 8 h, 12 h, and 24 h) and the fluorescence signal of Cy7 was detected through the fluorescence spectrophotometer (PerkinElmer, USA) and calculated according to the standard curve. Pharmacokinetic parameters were calculated by DAS 2.1.1 software. For the detection of biodistribution, IL-15-Cy7 and biNV-IL-15-Cy7 were intravenously injected into BALB/c mice. At 6 h post-administration, the axillary and inguinal LNs and major organs (heart, liver, spleen, lung, and kidney) were collected for ex vivo fluorescence imaging. The organs were then homogenized in lysis buffer. The biodistribution of IL-15-Cy7 and biNV-IL-15-Cy7 was investigated via detecting the fluorescence intensity of Cy7 in organ lysate through a fluorescence spectrophotometer (PerkinElmer, USA) and calculated according to the standard curve.

## The improved therapeutic window of IL-15 therapy

Female BALB/c mice aged 6–8 weeks were applied and acclimatized to the animal center. The in vivo safety of IL-15 and biNV-IL-15 was evaluated with normal BALB/c mice. As indicated, IL-15 and biNV-IL-15 were intravenously administrated with multiple doses on days 0, 3, 6, and 9. Body weight change was recorded every 2 d. Once the mice became moribund or the body weight losses were more than 20% of the initial weight, the mice were euthanized. In addition, the blood sample was taken on day 14 for hematologic and biochemical analysis including red blood cell count (RBC), platelet count (PLT), aspartate transaminase (AST), alanine transaminase (ALT), creatinine (CREA), and alkaline phosphatase (ALP).

## In vivo antitumor activity

$1 \times 10^6$ 4T1, CT26 or B16F10-OVA cancer cells were subcutaneously inoculated into BALB/c or C57BL/6j mice (female, 6–8 weeks old) and divided into 6 groups randomly. On days 7, 10, 13, and 16, the PBS, DC vesicles, IL-15, biNV, IL-15+biNV, and biNV-IL-15 (60 µg DC vesicles or 3 µg IL-15) were intravenously administrated into the mice. Furthermore, the 4T1 tumor model was also established to compare the therapeutic outcomes of heterodimeric IL-15 and the MIST platform. Mice were divided into 4 groups randomly. On days 7, 10, 13, and 16, the PBS, IL-15:IL-15Rα, IL-15:IL-15Rα+biNV, and biNV-IL-15 (60 µg DC vesicles or 3 µg IL-15) were intravenously administrated into the mice. 4T1 or CT26 tumor-derived HCP and OVA served as the tumor antigens of 4T1, CT26 and B16F10-OVA, respectively. The tumor size was monitored every 2 d following the formula below: 1/2 × length × width$^2$, and the survival rate was observed daily. To confirm the safety of various treatments, the heart, liver, spleen, lung, and kidney were harvested from one mouse per group in 4T1 and CT26 tumor models for hematoxylin and eosin (H&E) staining.

To prove the long-lasting anticancer immunological memory activated by biNV-IL-15, the mice that survived in the biNV-IL-15 group on day 55 were rechallenged with $1 \times 10^6$ 4T1, CT26, and B16F10-OVA cancer cells, respectively. As a control, naive mice of the same age were individually implanted with 4T1, CT26, or B16F10-OVA cancer cells. Tumor volume and survival rate were recorded. For in vivo lymphocyte depletion, 100 µg anti-CD4 antibody (BioXCell, clone GK1.5) or 100 µg

anti-CD8a antibody (BioXCell, clone 2.43) were intraperitoneally injected into the mice on days 4, 7, 10, 13, and 16, respectively. To evaluate the in vivo antigen-specific cancer inhibition of biNV-IL-15, $1 \times 10^6$ 4T1 or B16F10-OVA cancer cells were subcutaneously transplanted into BALB/c or C57BL/6j mice. BMDCs harvested from BALB/c and C57BL/6j mice were separately used to prepare DC vesicles for 4T1 and B16F10-OVA treatment. 4T1 tumor-derived HCP and OVA served as the tumor antigens of 4T1 and B16F10-OVA. On days 7, 10, 13, and 16, the PBS, biNV-IL-15/HCP, and biNV-IL-15/OVA (60 μg DC vesicles per mouse) were intravenously administrated into the mice, respectively. Tumor volume and survival rate were recorded.

## The suppression of tumor recurrence

To explore the capacity in inhibiting postsurgical tumor regrowth of biNV-IL-15, $1 \times 10^6$ 4T1-luc cancer cells were transplanted into the right mammary fat pads of female BALB/c mice aged 6–8 weeks. Mice were divided into 6 groups randomly on day 10, the tumor was resected with ~1% leaving to simulate residual microtumor after surgery. On days 10, 13, 16, and 19, the PBS, DC vesicles, IL-15, biNV, IL-15+biNV, and biNV-IL-15 (60 μg DC vesicles or 3 μg IL-15) were intravenously administrated into the mice, respectively. Furthermore, the tumor resection model was also established to compare the anti-recurrence effects of heterodimeric IL-15 and the MIST platform. Mice were divided into 4 groups randomly on day 10, the tumor was resected with ~1% leaving to simulate residual microtumor after surgery. On days 10, 13, 16, and 19, the PBS, IL-15:IL-15Rα, IL-15:IL-15Rα+biNV, and biNV-IL-15 (60 μg DC vesicles or 3 μg IL-15) were intravenously administrated into the mice, respectively. The recurrent tumor volume was determined using a caliper every 2 d, and the survival rate was recorded for 55 d. On days 10, 11, 17, and 22, in vivo imaging system (IVIS) (PerkinElmer, USA) was used to observe the bioluminescence of the tumor at 10 min post intraperitoneal administration of 3 mg D-Luciferin (Dalian Meilun Biotechnology Co., Ltd., China) per mouse. Living Image (PerkinElmer, USA) software was applied for data analysis. Images on day 10 were shown before surgery.

## In vivo anti-metastasis performance

4T1-luc cancer cells ($1 \times 10^6$ cells) were orthotopically implanted into the left mammary fat pads of female BALB/c mice aged 6-8 weeks. Tumor-bearing mice were divided into 6 groups randomly. PBS, DC vesicles, IL-15, biNV, IL-15+biNV, and biNV-IL-15 (60 μg DC vesicles or 3 μg IL-15) were administrated individually on days 7, 10, 13, and 16. Moreover, the hematogenous metastasis model was also established to compare the antimetastatic ability of heterodimeric IL-15 and the MIST platform. Tumor-bearing mice were divided into 4 groups randomly. On days 7, 10, 13, and 16, the PBS, IL-15:IL-15Rα, IL-15:IL-15Rα+biNV, and biNV-IL-15 (60 μg DC vesicles or 3 μg IL-15) were intravenously administrated into the mice. To simulate a more malignant invasion and hematogenous metastasis, we intravenously injected $5 \times 10^5$ 4T1-luc cells into the mice on day 10. On day 20, fresh lungs were harvested from mice after euthanasia, and incubated in D-Luciferin (15 mg mL$^{-1}$) for 10 min. Cancer metastasis in the lung was monitored by ex vivo bioluminescent imaging. Meanwhile, the lung was stained in Bouin's fluid and H&E staining for lung and liver slices was also conducted to evaluate the anti-metastasis efficiency. The metastatic area in slices was calculated with ImageJ software (v1.8.0). Tumor size and survival rate were recorded.

## Flow cytometry

To assess the DC maturation, immature DCs and DC-IL-15/HCP were stained with fluorescence-labeled antibodies CD11c (Biolegend, cat. no. 117306, Clone: N418, 1:200 dilution), CD80 (Biolegend, cat. no. 104707, Clone:16-10A1, 1:200 dilution), CD86 (Biolegend, cat. no. 105011, Clone:GL-1, 1:200 dilution) for flow cytometry assay. As for in vivo flow cytometric analysis, tumors isolated from mice were

separated into small sections. The single-cell suspension was obtained from sections through homogenization in staining buffer containing digestive enzyme. Then cell was stained using fluorescence-labeled antibodies CD11b (Biolegend, cat. no. 101208, Clone:M1/70, 1:200 dilution), F4/80 (Biolegend, cat. no. 123116, Clone:BM8, 1:200 dilution), CD80 (Biolegend, cat. no. 104722, Clone:16-10A1, 1:200 dilution), CD206 (Biolegend, cat. no. 141716, Clone:C068C2, 1:200 dilution) to identify tumor-associated macrophages. As for tumor-infiltrating T cells, fluorescence-labeled antibodies CD3 (Biolegend, cat. no. 100204, Clone:17A2, 1:200 dilution), CD4 (Biolegend, cat. no. 100432, Clone:GK1.5, 1:200 dilution), CD8 (Biolegend, cat. no. 100712, Clone:53-6.7, 1:200 dilution), Foxp3 (Biolegend, cat. no. 126404, Clone:MF-14, 1:200 dilution) were applied according to the instruction supplied by the manufacturer. For intracellular staining of Foxp3, cells were fixed and permeabilized using a Foxp3 Staining Buffer Set (eBioscience) per the manufacturer's procedure. Finally, flow cytometer (BD FACSCalibur) was applied to measure the stained cells and FlowJo software was employed to analyze the data.

The isolation of draining lymph nodes (dLN) was performed with surgical equipment, followed by crushing between the surface of frosted microscope slides into a well containing PBS. The cell mixture was then filtered using a 70 μm cell strainer into the conical tube. After filtration, stained the cells using fluorescence-labeled antibodies CD3 (Biolegend, cat. no. 100204, Clone:17A2, 1:200 dilution), CD4 (Biolegend, cat. no. 100432, Clone:GK1.5, 1:200 dilution), CD8 (Biolegend, cat. no. 100712, Clone:53-6.7, 1:200 dilution), Foxp3 (Biolegend, cat. no. 126404, Clone:MF-14, 1:200 dilution) referring to the manufacturer's protocol. For intracellular staining of Foxp3, cells were fixed and permeabilized using a Foxp3 Staining Buffer Set (eBioscience) per the manufacturer's procedure. The spleen was surgically extracted through sterilized surgical equipment. The spleen mixture was filtered into the conical tube through a filter, and centrifuged for 5 min. After washing the mixture, the cell pellet was suspended using red blood cell lysis buffer for 5 min. Cells were stained with fluorescence-labeled antibodies CD3 (Biolegend, cat. no. 100218, Clone:17A2, 1:200 dilution), CD62L (Biolegend, cat. no. 104428, Clone:MEL-14, 1:200 dilution), CD8 (Biolegend, cat. no. 100706, Clone:53-6.7, 1:200 dilution), and CD44 (Biolegend, cat. no. 103008, Clone: IM7, 1: 200 dilution) according to the manufacturer's protocol. Stained cells were monitored using flow cytometer (BD FACSCalibur) and evaluated using FlowJo software.

The tetramer staining analysis through peptide-MHC tetramer tagged using PE (H-2K$^b$-restricted SIINFEKL, eBioscience, cat. no. 12-5743-81, Clone:25-D1.16, 1:400 dilution) was performed to investigate the percentage of antigen-specific CD8$^+$ T cell in the B16F10-OVA tumor model. In brief, resuspended peripheral blood lymphocyte with the solution of 0.025 mg mL$^{-1}$ mouse CD16/32 antibody for blocking FcR-mediated and non-specific antibody binding. Then, incubated the suspension for 10 min at 25 °C and washed it 5 times using FACS buffer. Subsequently, added H-2K$^b$ OVA Tetramer-SIINFEKL-PE solution into the samples and incubated for 30 min on ice. Next, added anti-CD8-APC (Biolegend, cat. no. 100712, Clone:53-6.7, 1:200 dilution) for 20 min incubation on ice. Afterwards, washed the samples twice using FACS buffer, and fixed the samples. Stained cells were analyzed with flow cytometer (BD FACSCalibur) and evaluated using FlowJo software.

## Immunohistochemistry assay

Tumor tissue was fixed with 4% paraformaldehyde, the paraffin-embedded section was then deparaffinized, dehydrated, treated using 3% $H_2O_2$ for blocking endogenous peroxidase, and incubated with serum to prevent non-specific antibody binding and reduce background and prevent false positive results. Afterwards, incubated the slide with primary antibodies of anti-HSP70 (Santa Cruz, sc-24, Clone:W27, 1:200 dilution) and anti-TNF-α (Abcam, ab1793, Clone:52B83, 1:40 dilution), respectively. Following washes, incubated the slide with

biotinylated secondary antibody. The slide was next treated with streptavidin-peroxidase complex and 3,3'-diaminobenzidine chromogen solution (DAB, DakoCytomation), and counterstained using Mayer's hematoxylin. The Olympus BX60 light microscope was applied to visualize slide.

### Immunofluorescence staining

Tumor tissue was harvested from mice and snap-frozen utilizing an optimal cutting temperature compound. Then, cut the tumor tissue with a cryotome, mounted on a slide, and incubated using primary antibodies F4/80 (Abcam, cat. no. ab100790, 1:200 dilution) and CD8 (Abcam, cat. no. ab22378, 1:200 dilution) at 4 °C overnight. After fluorescence-labeled secondary antibodies (goat anti-rabbit IgG (H + L; ThermoFisher, cat. no. A32733, 1:1000 dilution) and goat anti-rat IgG (H + L; ThermoFisher, cat. no. A18866, 1:600 dilution)) addition, the samples were observed by CLSM (C2SI, Nikon, Japan). The antibody applied in this experiment was diluted 200 times.

### Enzyme-linked immunosorbent assay (ELISA)

The IFN-γ secretion in the tumor microenvironment was investigated through ELISA analysis. Tumor tissue harvested from tumor-bearing mice was homogenized in PBS at 4 °C. The sample was centrifuged at $2500 \times g$ for 10 min, and the supernatant was gathered and tested either for IFN-γ with an ELISA kit or total protein with a bicinchoninic acid (BCA) kit referring to the manufacturer's protocol. The concentrations of IFN-γ were normalized to total protein.

### Statistical analysis

Data are shown as mean ± s.d. Survival benefit was examined with log-rank test, and two-group comparison was conducted with one-way analysis of variance (ANOVA) and Student's t-test. All statistical analyses were performed with the GraphPad prism 8.0. $P < 0.05$ was determined as the statistical significance threshold.

### Reporting summary

Further information on research design is available in the Nature Portfolio Reporting Summary linked to this article.

## Data availability

All data supporting the findings of this study are available within the Article, Supplementary Information or Source Data file. The source data underlying Figs. 2g, i, 3c–i, k, 4b–h, 5b–f, h, j, 6b, d–f, 7c–e, h, j, l, n, p, 8c–e, g, i, k, m, o, Supplementary Figs. 2, 3, 4, 7, 8b–g, i, k, l, 9b–e, 12, 16a, b, 14a, b, 15b, c, 17c, d, f, h, j, l, n, 21b, 23, 24c–e, g, i, k, m, o have been deposited in the Figshare database (https://doi.org/10.6084/m9.figshare.24079164)[57]. Source Data.xlsx (figshare.com) Source data are provided with this paper.

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

## Acknowledgements

This research was supported by National Key R&D Program of China (2021YFA0909900 (J.S.) and 2022YFE0111600 (J.S.)), National Natural Science Foundation of China (No. 82073777 (J.S.), 82273874 (J.S.), and 82002253 (Z.F.)), the National University of Singapore Start-up Grant (NUHSRO/2020/133/Startup/08 (X.C.), NUHSRO/2022/005/Startup/02 (X.C.)), The Ministry of Education (MOE) Academic Research Fund (AcRF) Tier 1 (NUHSRO/2022/068/T1/Seed-Mar/04 (X.C.)), NUS School of Medicine Nanomedicine Translational Research Programme (NUHSRO/2021/034/TRP/09/Nanomedicine (X.C.)), National Medical Research Council (NMRC) Centre Grant Programme (CG21APR1005 (X.C.)), and Singapore Ministry of Education, Academic Research Fund Tier 2 (T2EP30122-0002 (X.C.)). K.W. acknowledges the China Scholarship Council (CSC) for financially supporting his work at the National University of Singapore (No. 202208210238 (K.W.)). We thank Liyan Li and Haocai Chang for their invaluable help and advices.

## Author contributions

All authors have given approval to the final version of the manuscript. K.W., Z.H., X.C. and J.S. conceived the project and designed experiments; K.W., X.Z., H.Y., X.W., Z.F., Q.L., S.L. and J.Z. performed research; K.W. and X.Z. analyzed data; S.Z. provided useful suggestions; K.W., Z.H., Q.N., X.C. and J.S. wrote the initial draft. All authors contributed to the writing of the final manuscript.

## Competing interests

The authors declare no competing interests.
