## [Peer Review File · Nature Communications]

Reviewers' Comments:

Reviewer #1:

Remarks to the Author:

The paper by Wang and colleagues examines the biology and efficacy of a biomimetic nanovaccine in which IL-15 superagonist, TAA/MHCI and co-stimulatory molecules are anchored to DC-derived cytomembrane vesicles. The authors report that the nanovaccine has a longer plasma half-life, elicits potent T cell responses, has minimal in vivo toxicity and is efficacious against several syngeneic tumor models. The authors should be complemented on developing a complex cell-based immunotherapeutic that harnesses many of the beneficial effects of IL-15 while minimizing toxicity in mice. Some of the the hurdles for translation to humans are the current use of an adenovirus in the construct and the complex manufacturing process. Nevertheless, the authors provide good proof of concept data in mice. The following comments are offered.

1. The efficacy and safety of the construct is quite good based on the presented data. The authors use free IL-15 as a major control in their studies. IL-15 superagonist (IL-15/IL-15 α) is more potent and is considered to be more favorable from an antitumor efficacy and drug development perspective. Some comment on their choice of IL-15 as a control is warranted.
2. Body weight and organ injury are good endpoints for assessing toxicity. Splenomegaly and increased plasma cytokines are common problems in mice treated with IL-15 superagonist. Did the authors assess spleen size and plasma cytokines in their study?

Reviewer #2:

Remarks to the Author:

The manuscript NCOMMS-23-00992-T "Biomimetic nanovaccine-mediated multivalent IL-15 self-transpresentation for potent and safe cancer immunotherapy" presents the results of a complex and preliminary study of the feasibility and effectiveness of a novel anti-cancer dendritic cell (DC)-derived nano-vesicle vaccine with multiple immune functions.

DC engineered to express IL-15/IL-15Ra membrane molecules, then pulsed with Heat Shock Protein -70 complexed with tumor proteins (HCP) to transpresentation of tumor antigens through MHC I (and co-receptors) to CD8+ Cytotoxic T Lymphocytes (CTL) are treated to prepare nano-vesicles coated with all necessary molecules for an intense CTL priming and activation, including the generation of memory cells.

The goal is ambitious and the study was conducted with a very extensive and logic design.

The work is an original upgrading of an approach to using the DC as functional membranes (see below) that, engineered and expressing appropriate antigen on MHC I molecule can efficiently and with the highest biocompatibility activate a strong CTL response. A base of the originality is linking the effect of IL-15 to the presentation of the tumor antigen in a multivalent system.

The study indicates the feasibility of the hypothesis and it is significant in the hot field of immunotherapy of cancer, due to its very perspective possibilities of translation in medicine.

All the steps were coherently followed, proving with adequate sequence and precision the various milestones of the research, clearly described and sustained by the presented documentation.

The results are adequately presented and discussed. The materials in the main text and in the Supplementary part (figures and schemes) are abundant but not redundant and clearly support the descriptions.

The presented and discussed data support the achievement of a model of treatment that, in this first study, confirms with good evidence the hypothesis of multivalent and multimolecular immune stimulation of CTL using DC membranes. Under this aspect, a citation should be added in the selected literature, because in part it anticipates the efficient approach here demonstrated:

Kovar M, Boyman O, Shen X, Hwang I, Kohler R, Sprent J. Direct stimulation of T cells by membrane vesicles from antigen-presenting cells. Proc Natl Acad Sci U S A. 2006 Aug 1;103(31):11671-6. doi: 10.1073/pnas.0603466103. Epub 2006 Jul 19. PMID: 16855047; PMCID: PMC1544228.

The manuscript needs some upgrades and minor revisions:

Fig.1a should have a clearer design and description because, as it is now, looks that IL-15 appears because of the antigen pulse, while in the description is understood that before the DC is engineered to express IL-15 then it is pulsed with HCP to present the tumor antigen on MHC I complex.

In Fig. 6a, due to the dimension of the mice, is not immediate to recognize what are the symbols related to the syringes (cells or therapy), and the last mouse (for imaging) doesn't show the orthotopic tumor, only metastatic lungs.

The texts on page 5 lines 77-82 and 88-90 should be put together and remodeled to avoid repetitions.

Since HSP-70 is highly represented in 4T1 and CT26 cancer cells, why not obtain it directly from cultured cell lines instead of using cells dissociated from tumors developed in animals? An in vivo tumor is not homogeneous as in the cultured cell line due to the components of the tissue in which the cells are inoculated and developed. The presence of other cells positive to HSP-70, also in the tissue indicated as NT in the images, increases the possibility of a not fully pure HSP-70-Ag (HCP) of only tumor origin (even though prevalent).

In preliminary experiments, were the HCP alone co-incubated with CD8+ cells to see their potential to directly prime CTL in vitro? If it was done, it should be fine as a note in the Discussion. At page 7 line 133 do you mean "50µg HCP/l /g tumor tissue"?

Please, revise the "Supplementary and additional material" part because suppliers of materials and type/company of used instruments (ELISPOT reader; Spectrophotometer; Cytotoxicity Detection Kit; Exosome Spin Column; D-Luciferin; in vivo imaging system; imaging analysis software, etc.) are not always cited.

In Animal studies (Supplementary, 143-146), the source, age and rearing conditions of animals are not indicated.

In Flow cytometry (lines 220-226) should be indicated that the described procedure is for identifying the TILs. Was CD45, used to gate immune cells? To stain Foxp3 is necessary permeabilization of the cells, but here it is not described; please add this passage and the method that you used.

In the Immunohistochemistry assay (lines 252-259) the block of endogenous peroxidases is indicated but not the inhibition of the block to prevent non-specific antibody binding and reduce background and prevent false positive results using sera or a protein. Please, show it, if it was performed.

The indication of time is sometimes variable, either in minutes or in fractions (e.g. 0.03) of the hour. To use only one modality is recommended.

Why to produce an orthotopic tumor and metastases in the same animal if the aim of looking for anti-tumor therapy was already done? Generally, the orthotopic implant of tumor cells is used to study metastasis by following the spontaneous spread from the primary tumor (it takes time). The i.v. (in the tail veins – never cited in the text) injection of cells for lung colonization is a quicker simulation of the metastatic process (very used, even though biologically not overlap the natural process). Two pathologic conditions together may influence each other and increase animal suffering. What was the reason of this choice? It should be indicated.

Some revision of English might be useful, especially in Methods.

Reviewer #3:

Remarks to the Author:

In this manuscript, Wang et al. developed a biomimetic nanovaccine with multivalent IL-15 self-transpresentation (MIST) using cytomembrane vesicles from dendritic cells infected by adenovirus expressing IL15 and IL15Ralpha, together with tumor antigens. They characterized the nanovaccine platform in size, in vivo distribution, biosafety, and most importantly, antitumor efficacy in multiple mouse tumor models. Overall I think it is a nice study with well-designed experiments and impressive results, but a bit short of biological insights. I have a few questions about this manuscript:

1. By using dendritic cell membrane as the surrogate source of antigen-MHC complex to stimulate T cells has been reported previously by the same group and other groups, e.g. as in references 23 and 30. The authors may want to explicitly describe how this current study differ from the previous studies using similar strategy for T cell stimulation besides IL15 transpresntation.

2. How would the authors define the cytomembrane vesicles? Do they contain membrane debris or exosomes? Or are they equivalent to the exosomes?

3. In the results description for tumor immune microenvironment analysis in Fig. 4C, the authors suggested that MIST treatment converted M2 to M1 macrophages, but in Fig. 5F and S5, S9, the infiltration number of F4/80+ cells increased as well as CD8+ cells. F4/80 is generally considered a pan marker for macrophage, why would the MIST treatment increase the total number of tumor-infiltrating macrophages?

4. The authors used recombinant IL15 as a control group. In the method part, the authors only mentioned that the recombinant IL15 was from Abcam Inc. without Cat # information. Is the recombinant IL15 is the monomer cytokine only, or the IL15-IL15Ralpha heterodimer complex as the authors expressed in dendritic cells? This is important, as the latter can better serve as a control group for the MIST treatment group. For example, NIZ985, a recombinant heterodimer of IL15-IL15Ralpha, are currently under clinical trials for cancer therapy. So in my opinion the MIST platform, but not the transpresentation, is the major point of this manuscript.

56. Minor points: In the introduction part, page 4, line 56-58, the authors wrote: "IL-15 supports the generation and survival of memory T cells, while IL-2 promotes immunosuppressive regulatory T cell (Treg) expansion". This is misleading, as IL2 is a T cell survival factor for all T cell subsets including effector T cells and Tregs. In another sentence, the authors claimed "In addition, the distinct size of IL-15 is below the renal filtration threshold (<40 kDa), resulting in rapid clearance and short-lived effects." This is partially true, but it is not the main reason for the limit of IL15 clinical use. For example, IFNalpha and IL2 are also small proteins, but both have been approved by FDA for cancer treatment. Please revise these sentences to accurately describe the current limit of clinical use of cytokines in cancer therapy.

Manuscript ID: NCOMMS-23-00992-T

Title: Biomimetic nanovaccine-mediated multivalent IL-15 self-transpresentation (MIST) for potent and safe cancer immunotherapy

Dear reviewers,

We are truly grateful for your constructive comments. We have made substantial changes. The revisions are highlighted in red. We also provide below a point-by-point response to the comments.

We appreciate the critical reviews of the manuscript and hope the revised version has been improved to a significant degree to merit further consideration for publication.

Response to Reviewers' comments:

Reviewer #1:

The paper by Wang and colleagues examines the biology and efficacy of a biomimetic nanovaccine in which IL-15 superagonist, TAA/MHCI and co-stimulatory molecules are anchored to DC-derived cytomembrane vesicles. The authors report that the nanovaccine has a longer plasma half-life, elicits potent T cell responses, has minimal in vivo toxicity and is efficacious against several syngeneic tumor models. The authors should be complemented on developing a complex cell-based immunotherapeutic that harnesses many of the beneficial effects of IL-15 while minimizing toxicity in mice. Some of the hurdles for translation to humans are the current use of an adenovirus in the construct and the complex manufacturing process. Nevertheless, the authors provide good proof of concept data in mice. The following comments are offered.

1. The efficacy and safety of the construct is quite good based on the presented data. The authors use free IL-15 as a major control in their studies. IL-15 superagonist (IL-15/IL-15 α) is more potent and is considered to be more favorable from an antitumor efficacy and drug development perspective. Some comment on their choice of IL-15 as a control is warranted.

Response: We appreciate the reviewer's comments. To further evaluate the antitumor efficacy of biNV-IL-15, we constructed a heterodimer cytokine (IL-15 superagonist, IL-15:IL-15 α) consisting of IL-15 and IL-15 α , which was produced and purified from a cloned cell line derived from the HEK293 cell line transfected with optimized plasmid DNA encoding IL-15 and IL-15 α (*J Immunother Cancer*. 2020, 8(1), e000599; *J Biol Chem*. 2013, 288(25), 18093-18103). We then compared the therapeutic outcomes of IL-15:IL-15 α , IL-15:IL-15 α +biNV, and biNV-IL-15 in the

4T1 tumor model, tumor resection model, and hematogenous metastasis model. In contrast to IL-15:IL-15R α and IL-15:IL-15R α +biNV, biNV-IL-15 exhibited superior antitumor activity, anti-recurrence effect, and metastasis-inhibitory ability, demonstrating the potential for practical clinical application. The immune responses after various treatments were also explored through flow cytometric assays. The results indicated that biNV-IL-15 treatment effectively activated antitumor immunity including improved proportion of CD8⁺ T cells, M1-type macrophage polarization, and increased T_{EM} percentage, demonstrating significant advantages over IL-15:IL-15R α and IL-15:IL-15R α +biNV.

The results were discussed in the revised manuscript : “Furthermore, the 4T1 tumor model was also established to compare the therapeutic outcomes of heterodimeric IL-15 (IL-15:IL-15R α) and the MIST platform (Supplementary Fig. 9a). As shown in Supplementary Fig. 9b, the biNV-IL-15 exhibited superior antitumor activity compared with IL-15:IL-15R α and IL-15:IL-15R α +biNV, leading to prolonged survival period of mice”; “The flow cytometric assays were also performed in 4T1 tumor model established to compare the efficacy of heterodimeric IL-15 and the MIST platform. As shown in Supplementary Fig. 9c, e, biNV-IL-15 effectively enhanced the proportion of CD8⁺ T cells and decreased the immunosuppressive Tregs in tumors and dLNs, displaying obvious advantages over IL-15:IL-15R α and IL-15:IL-15R α +biNV. Meanwhile, biNV-IL-15 significantly remodeled TAMs from M2 type to M1 type and upregulated T_{EM} level in the spleen in comparison with IL-15:IL-15R α and IL-15:IL-15R α +biNV (Supplementary Fig. 9c, d)”; “Furthermore, the tumor resection model was also established to compare the anti-recurrence effects of heterodimeric IL-15 and the MIST platform (Supplementary Fig. 16a). As shown in Supplementary Fig. 16b-d, the biNV-IL-15 displayed the excellent capability of tumor recurrence inhibition in contrast to IL-15:IL-15R α and IL-15:IL-15R α +biNV, resulting in superior survival outcomes”; “The flow cytometric assays were also performed in tumor resection model established to compare the efficacy of heterodimeric IL-15 and the MIST platform. As shown in Supplementary Fig. 16e-h, biNV-IL-15 obviously improved the percentage of tumor-infiltrating CD8⁺ T cells and downregulated the level of immunosuppressive Tregs in tumor, implying remarkable advantages over IL-15:IL-15R α and IL-15:IL-15R α +biNV. Furthermore, biNV-IL-15 effectively polarized M2-like TAMs to M1-like TAMs and increased splenic T_{EM} proportion in contrast to IL-15:IL-15R α and IL-15:IL-15R α +biNV (Supplementary Fig. 16i-n)”; “Moreover, the hematogenous metastasis model was also established to compare the antimetastatic ability of heterodimeric IL-15 and the MIST platform (Supplementary Fig. 19a). As shown in Supplementary Fig. 19b-k, the biNV-IL-15 treatment not only inhibited the tumor growth but also prevented lung and liver metastasis of the tumor in contrast to IL-15:IL-15R α and IL-15:IL-15R α +biNV, causing an extended survival time”; “The flow cytometric assays were also

performed in hematogenous metastasis model established to compare the metastasis-inhibitory ability of heterodimeric IL-15 and the MIST platform. As shown in Supplementary Fig. 19l-o, biNV-IL-15 significantly elevated the level of CD8⁺ T cells and decreased the proportion of Tregs in blood, demonstrating distinctive advantages over IL-15:IL-15R α and IL-15:IL-15R α +biNV”.

Supplementary Fig. 9. *In vivo* antitumor activity of biomimetic nanovaccine-mediated multivalent IL-15 self-transpresentation (MIST). (a) 4T1 cancer cells were subcutaneously inoculated into BALB/c mice. On days 7, 10, 13, and 16, the PBS, IL-15:IL-15R α , IL-15:IL-15R α +biNV, and biNV-IL-15 were intravenously administered into the mice. (b) Average tumor growth curve and survival curve following various treatments (n = 6/group). (c) Flow cytometric quantification of tumor-infiltrating CD8⁺ T cells, CD4⁺Foxp3⁺ Tregs, M1-type TAMs (CD80^{hi}CD11b⁺F4/80⁺), and M2-type TAMs (CD206^{hi}CD11b⁺F4/80⁺) in 4T1 tumor model following various treatments (n = 4/group). (d) Flow cytometric quantification of CD3⁺CD8⁺CD62L^{low}CD44^{hi} effector memory T cells (T_{EM}) in the spleen after various treatments (n = 4/group). (e) Flow cytometric quantification of CD8⁺ T cells and CD4⁺Foxp3⁺ Tregs in dLNs (n = 4/group). Data represent the mean \pm s.d. Statistical significance was calculated through two-tailed student’s t-test (b), log-rank (Mantel-Cox) test (b), or one-way ANOVA using a Tukey post-hoc test (c-e).

Supplementary Fig. 16. The suppression of tumor recurrence. (a) Schematic depicting biNV-IL-15

treatment in 4T1-luc orthotopic cancer model with incomplete cancer resection (IVIS: *in vivo* imaging system; FC: flow cytometry analysis). (b) *In vivo* bioluminescence imaging for 4T1-luc tumor following the removal of the primary tumor. Every group showed four representative mice. Images on day 10 were shown before surgery. (c) Average tumor growth curve and (d) survival curve (n = 6/group) in tumor resection model receiving various treatments. (e) Flow cytometric examination images and (f) relative quantification of CD8⁺ T cell in tumor (n = 4/group). (g) Flow cytometric measurement images and (h) relative quantification of CD4⁺Foxp3⁺ Tregs in tumor (n = 4/group). (i) Flow cytometric assessment images and (j) relative quantification of M1-type TAMs (CD80^{hi}CD11b⁺F4/80⁺) in tumor (n = 4/group). (k) Flow cytometric analysis images and (l) relative quantification of M2-type TAM (CD206^{hi}CD11b⁺F4/80⁺) in tumor (n = 4/group). (m) Flow cytometric evaluation images and (n) relative quantification of CD3⁺CD8⁺CD62L^{low}CD44^{hi} T_{EM} in the spleen (n = 4/group). Data represent the mean ± s.d. Statistical significance was calculated through two-tailed student's t-test (c), log-rank (Mantel-Cox) test (d), or one-way ANOVA using a Tukey post-hoc test (f, h, j, l, n).

Supplementary Fig. 19. *In vivo* anti-metastasis performance. (a) 4T1-luc cancer cells were intravenously injected into the tumor-bearing mice on day 10 to simulate the hematogenous metastasis model (FC: flow cytometry analysis; H&E staining: hematoxylin and eosin staining). PBS, IL-15:IL-15R α , IL-15:IL-15R α +biNV, and biNV-IL-15 were individually administered on days 7, 10, 13, and 16. (b) *Ex vivo* bioluminescent imaging and (c) average bioluminescent radiance for isolated lungs were studied after various treatments on day 20. (d) Average tumor growth curve and (e) survival curve (n = 6/group) in hematogenous metastasis model receiving various treatments. (f) Graphs and (g) quantification of lung metastases (n = 5/group) for Bouin's fluid staining of lungs. H&E staining of the lung (h) and liver (j) slices after different treatments. Scale bar = 100 μ m. Quantification for metastasis area percentages of the lung (i) and liver (k) slices (n = 3/group). (l) Flow cytometric analysis images and (m) relative quantification of CD8⁺ T cells in the blood (n = 4/group). (n) Flow cytometric examination images and (o) relative quantification of CD4⁺Foxp3⁺ Tregs in the blood (n = 4/group). Data represent the mean \pm s.d. Statistical significance was calculated through one-way ANOVA using a Tukey post-hoc test (c, g, i, k, m, o), two-tailed student's t-test (d) or log-rank (Mantel-Cox) test (e).

2. Body weight and organ injury are good endpoints for assessing toxicity. Splenomegaly and increased plasma cytokines are common problems in mice treated with IL-15 superagonist. Did the authors assess spleen size and plasma cytokines in their study?

Response: We appreciate the reviewer's comments. As reviewer suggested, we further assessed spleen size and serum inflammatory cytokines (CXCL10 and IL-6) of mice treated with biNV-IL-15 and IL-15 superagonist (IL-15:IL-15R α). Compared with biNV-IL-15, IL-15:IL-15R α induced an obvious splenomegaly and significantly increased serum levels of inflammatory cytokines such as CXC motif chemokine ligand 10 (CXCL10) and interleukin-6 (IL-6) (Fig. R1), confirming the superior safety of the MIST platform.

Figure R1. The assessment of spleen size and serum inflammatory cytokines after biNV-IL-15 administration. (a) Photographs and (b) quantification of spleen/body weight ratios of mice after treatment with PBS, biNV-IL-15, and IL-15:IL-15R α on days 0 and 3 (n = 5/group). For the detection of serum inflammatory cytokines, BALB/c mice were treated with PBS, biNV-IL-15, and IL-15:IL-15R α on days 0 and 3. Blood was collected at 12 h after the last treatment. Levels of (c) CXCL10 and (d) IL-6 in the serum were measured (n = 5/group). Data represent the mean \pm s.d. Statistical significance was calculated through one-way ANOVA using a Tukey post-hoc test (b-d).

Reviewer #2:

The manuscript NCOMMS-23-00992-T “Biomimetic nanovaccine-mediated multivalent IL-15 self-transpresentation for potent and safe cancer immunotherapy” presents the results of a complex and preliminary study of the feasibility and effectiveness of a novel anti-cancer dendritic cell (DC)-derived nano-vesicle vaccine with multiple immune functions.

DC engineered to express IL-15/IL-15R α membrane molecules, then pulsed with Heat Shock Protein-70 complexed with tumor proteins (HCP) to transpresentation of tumor antigens through MHC I (and co-receptors) to CD8⁺ Cytotoxic T Lymphocytes (CTL) are treated to prepare nano-vesicles coated with all necessary molecules for an intense CTL priming and

activation, including the generation of memory cells.

The goal is ambitious and the study was conducted with a very extensive and logic design.

The work is an original upgrading of an approach to using the DC as functional membranes (see below) that, engineered and expressing appropriate antigen on MHC I molecule can efficiently and with the highest biocompatibility activate a strong CTL response. A base of the originality is linking the effect of IL-15 to the presentation of the tumor antigen in a multivalent system.

The study indicates the feasibility of the hypothesis and it is significant in the hot field of immunotherapy of cancer, due to its very perspective possibilities of translation in medicine.

All the steps were coherently followed, proving with adequate sequence and precision the various milestones of the research, clearly described and sustained by the presented documentation.

The results are adequately presented and discussed. The materials in the main text and in the Supplementary part (figures and schemes) are abundant but not redundant and clearly support the descriptions.

The presented and discussed data support the achievement of a model of treatment that, in this first study, confirms with good evidence the hypothesis of multivalent and multimolecular immune stimulation of CTL using DC membranes. Under this aspect, a citation should be added in the selected literature, because in part it anticipates the efficient approach here demonstrated:

Kovar M, Boyman O, Shen X, Hwang I, Kohler R, Sprent J. Direct stimulation of T cells by membrane vesicles from antigen-presenting cells. *Proc Natl Acad Sci U S A*. 2006 Aug 1;103(31):11671-6. doi: 10.1073/pnas.0603466103. Epub 2006 Jul 19. PMID: 16855047; PMCID: PMC1544228.

Response: We appreciate the reviewer's comments. As suggested, the reference has been added.

The manuscript needs some upgrades and minor revisions:

Fig.1a should have a clearer design and description because, as it is now, looks that IL-15 appears because of the antigen pulse, while in the description is understood that before the DC is engineered to express IL-15 then it is pulsed with HCP to present the tumor antigen on MHC I complex.

Response: We appreciate the reviewer's comments. As reviewer suggested, we have improved the

design and description of Fig.1a.

In Fig. 6a, due to the dimension of the mice, is not immediate to recognize what are the symbols related to the syringes (cells or therapy), and the last mouse (for imaging) doesn't show the orthotopic tumor, only metastatic lungs.

Response: We appreciate the reviewer's comments. As reviewer suggested, we have revised Fig. 6a to adjust the dimension of mice to easily recognize what are the symbols related to the syringes (cells or therapy) and show the orthotopic tumor in the last mouse (for imaging).

The texts on page 5 lines 77-82 and 88-90 should be put together and remodeled to avoid repetitions.

Response: We appreciate the reviewer's comments. As reviewer suggested, the texts on page 5 lines 77-82 and 88-90 have been put together and remodeled to avoid repetitions.

Since HSP-70 is highly represented in 4T1 and CT26 cancer cells, why not obtain it directly from cultured cell lines instead of using cells dissociated from tumors developed in animals? An in vivo tumor is not homogeneous as in the cultured cell line due to the components of the tissue in which the cells are inoculated and developed. The presence of other cells positive to HSP-70, also in the tissue indicated as NT in the images, increases the possibility of a not fully pure HSP-70-Ag (HCP) of only tumor origin (even though prevalent).

Response: We appreciate the reviewer's comments. Our primary goal is to translate this versatile and adaptable cytokine delivery platform into clinical practice. As part of our future preclinical development, we intend to collect tumor samples directly from patients. This approach of isolating HCP from tumor tissue, rather than relying on cultured cell lines or model antigens, holds great potential for clinical application. Meanwhile, the reliability of the procedure used to isolate HCP from tumor tissue has been validated in previous research (*Adv Sci.* 2020, 7(1), 1900069).

In preliminary experiments, were the HCP alone co-incubated with CD8⁺ cells to see their potential to directly prime CTL in vitro? If it was done, it should be fine as a note in the Discussion.

Response: We appreciate the reviewer's comments. The indispensable role of the interaction between the T cell receptor (TCR) and peptide/MHC molecules in T cell activation has been widely acknowledged in immunological theory (*Annu Rev Immunol.* 2002, 20(1), 621-667). Without processing and presentation by antigen-presenting cells (APCs), the antigen alone cannot directly

prime CTL *in vitro* (*J Exp Med.* 1990, 172(2), 631–640). In this study, the biomimetic nanovaccine derived from antigen-primed DCs comprises tumor-associated antigenic (TAA) peptide/MHC-I and costimulatory molecules to directly activate T cells.

At page 7 line 133 do you mean “50µg HCP/1 /g tumor tissue”?

Response: We appreciate the reviewer’s comments. It means that “50 µg of HCP was obtained per gram of tumor tissue”. To avoid misunderstanding, we have revised the manuscript.

Please, revise the “Supplementary and additional material” part because suppliers of materials and type/company of used instruments (ELISPOT reader; Spectrophotometer; Cytotoxicity Detection Kit; Exosome Spin Column; D-Luciferin; in vivo imaging system; imaging analysis software, etc.) are not always cited.

Response: We appreciate the reviewer’s comments. As reviewer suggested, we have revised the “Supplementary and additional material” part to cite suppliers of materials and type/company of used instruments.

In Animal studies (Supplementary, 143-146), the source, age and rearing conditions of animals are not indicated.

Response: We appreciate the reviewer’s comments. As reviewer suggested, the source, age and rearing conditions of animals have been indicated.

In Flow cytometry (lines 220-226) should be indicated that the described procedure is for identifying the TILs. Was CD45, used to gate immune cells? To stain Foxp3 is necessary permeabilization of the cells, but here it is not described; please add this passage and the method that you used.

Response: We appreciate the reviewer’s comments. As reviewer suggested, we have indicated in the revised manuscript that the described procedure is to identify the TILs. CD45 was not used to gate immune cells. This procedure was widely used in many articles (*J Am Chem Soc.* 2022, 144(2), 787-797; *ACS Nano.* 2023, <https://doi.org/10.1021/acsnano.3c01733>). For Foxp3 staining, the description of cell permeabilization has been added to the method.

In the Immunohistochemistry assay (lines 252-259) the block of endogenous peroxidases is indicated but not the indication of the block to prevent non-specific antibody binding and reduce background and prevent false positive results using sera or a protein. Please, show it, if

it was performed.

Response: We appreciate the reviewer's comments. As reviewer suggested, the description of blocking non-specific antibody binding using serum has been added to the method.

The indication of time is sometimes variable, either in minutes or in fractions (e.g. 0.03) of the hour. To use only one modality is recommended.

Response: We appreciate the reviewer's comments. As reviewer suggested, the indication of time has been revised into one modality.

Why to produce an orthotopic tumor and metastases in the same animal if the aim of looking for anti-tumor therapy was already done? Generally, the orthotopic implant of tumor cells is used to study metastasis by following the spontaneous spread from the primary tumor (it takes time). The i.v. (in the tail veins - never cited in the text) injection of cells for lung colonization is a quicker simulation of the metastatic process (very used, even though biologically not overlap the natural process). Two pathologic conditions together may influence each other and increase animal suffering. What was the reason of this choice? It should be indicated.

Response: We appreciate the reviewer's comments. As reviewer suggested, we explained in the main text why 4T1-luc tumor cells were additionally injected *i.v.* **“The additional tail vein injection of tumor cells into tumor-bearing mice to simulate hematogenous metastasis has been widely applied as an artificial whole-body spreading tumor model (*Nat Commun.* 2019, 10, 2025; *Nat Commun.* 2016, 7, 13193; *ACS Nano.* 2019, 13(5), 5662-5673; *Nano Today.* 2020, 35, 100987). Compared with spontaneous lung metastasis, the whole-body metastasis model was more aggressive and challenging, which was suitable for specialized anti-metastasis evaluation.”**

Some revision of English might be useful, especially in Methods.

Response: We appreciate the reviewer's comments. As reviewer suggested, we have carefully revised the English, especially in Methods.

Reviewer #3:

In this manuscript, Wang et al. developed a biomimetic nanovaccine with multivalent IL-15 self-transpresentation (MIST) using cytomembrane vesicles from dendritic cells infected by adenovirus expressing IL15 and IL15Ralpha, together with tumor antigens. They characterized the nanovaccine platform in size, in vivo distribution, biosafety, and most importantly, antitumor efficacy in multiple mouse tumor models. Overall, I think it is a nice study with well-

designed experiments and impressive results, but a bit short of biological insights. I have a few questions about this manuscript:

1. By using dendritic cell membrane as the surrogate source of antigen-MHC complex to stimulate T cells has been reported previously by the same group and other groups, e.g., as in references 23 and 30. The authors may want to explicitly describe how this current study differ from the previous studies using similar strategy for T cell stimulation besides IL15 transpresentation.

Response: We appreciate the reviewer's comments. By considering their hydrophilic cavity and hydrophobic bilayer structure, as well as their inherent biocompatibility, we sought to harness the exceptional features of biosynthetic cytomembrane vesicles for efficient therapeutic transportation. In our previous studies, we have successfully developed a range of biosynthetic cytomembrane vesicles that can serve as customizable shells for delivering bio-functional moieties and therapeutic entities (*Proc Natl Acad Sci.* 2015, 112(45), E6129-E6138; *Angewandte Chemie.* 2018, 130(38), 12679-12683; *Adv Mater.* 2019, 31(17), 1808294; *Nat Nanotechnol.* 2022, 17(5), 531-540). Our primary goal was to develop it as a versatile and adaptable model platform for future clinical applications. To achieve this, we have established a standardized operating procedure that ensures scaled synthesis and proper quality control of the cytomembrane vesicles.

In contrast to the traditional modification methods such as multivalent electrostatic interactions, receptor-ligand binding, and hydrophobic insertion, our genetic decorating procedure with desired proteins on the surface successfully hijacks the cellular activity and has several unique advantages: 1) the artificial proteins/peptides/domains can avoid undesired protein orientation, structural distortions and interfere with the existed membrane proteins because of cellular strict genetic control expression system; 2) the integrated proteins could have more post-translational modifications (e.g., glycosylation, acetylation, phosphorylation) that are critically important for their biological functions; 3) the existence of degeneracy and usage bias for genetic codes has excellent benefits for the same functional proteins expressing in different species but maintain their original sequence information and functions.

In this study, we synthesized genetically engineered biNV-IL-15 to enable multivalent IL-15 self-transpresentation. This approach resulted in extended blood circulation, an enhanced therapeutic window, and the ability to manipulate therapeutic T cells in a spatiotemporal manner, facilitating a broad spectrum antigen-specific T cell response with minimal systemic side effects. This platform technology demonstrates remarkable safety, effectiveness, and versatility, providing a promising solution to the challenges encountered in the clinical application of IL-15. Importantly, it introduces

a novel approach for the practical implementation of cytokine therapy in clinical settings.

In terms of using dendritic cell membrane as the surrogate source of antigen-MHC complex to stimulate T cells, we isolated primary bone marrow-derived DCs (BMDCs) for genetic engineering and antigen pulse in this study. In contrast, the DC2.4 cells used in references 23 and 30 are an immortalized cell line derived from C57BL/6 mice. Compared with primary dendritic cells, DC2.4 cells exhibit some functional differences and deficiencies (*J Pharm Pharmaceut Sci.* 2010, 13(1), 21-26). Moreover, the 4T1 and CT26 tumor models were derived from BALB/c mice, so the primary BMDCs from the same strain of mice are more reliable than DC2.4. In addition, we chose endogenous HSP70-chaperoned polypeptides (HCP) as tumor antigens to pulse BMDCs in this study instead of the commonly used model antigen ovalbumin (OVA). For many tumors, most endogenous tumor antigens are unknown. Studies have reported that HSP70 acts as a chaperone for several polypeptides that can generate tumor antigens, which can be represented by MHC molecules and elicit CD8⁺ T cell responses (*Science.* 1997, 278(5335), 117-120; *Annu Rev Immunol.* 2002, 20(1): 395-425). The immunogenic nature of the HCP allows for their novel use in personalized immunotherapy of cancer (*Adv Sci.* 2020, 7(1), 1900069). Our primary goal is to translate this versatile and adaptable cytokine delivery platform into clinical practice. As part of our future preclinical development, we intend to collect tumor samples directly from patients. This approach of isolating HCP from tumor tissue, rather than relying on cultured cell lines or model antigens, holds greater relevance for clinical application.

2. How would the authors define the cytomembrane vesicles? Do they contain membrane debris or exosomes? Or are they equivalent to the exosomes?

Response: We appreciate the reviewer's comments. The cytomembrane vesicles were different from exosomes and were isolated according to the following process (Fig. 1b). First, we gathered and washed the DCs twice in cold PBS containing protease inhibitor for removing cell debris. The DCs were then resuspended in PBS and sonicated in the sterile EP tube on ice. The cytomembrane vesicles were collected by multistep density gradient ultracentrifugation, and suspended in PBS. Finally, we introduced the isolated cytomembrane vesicles to the Mini-Extruder (Avanti Polar Lipids) with 200 nm pore-sized membrane filter in PBS to acquire uniform cytomembrane vesicles. The cytomembrane vesicles produced from the natural cell membrane have emerged as a promising alternative for biotherapy (*Nat Nanotechnol.* 2022, 17(5), 531-540; *Angewandte Chemie.* 2018, 130(38), 12679-12683; *Proc Natl Acad Sci.* 2015, 112(45), E6129-E6138). The inspiration for cytomembrane vesicles comes from the hijacking host cell machinery of enveloped viruses and the subsequent release (budding process) from the infected cells. Once the original cells are treated by ultrasonic crushing, cell membrane fragments will undergo a reassembly process to form

cytomembrane vesicles (*Adv Sci.* 2021, 8(21), 2100460).

As is well known to all, the exosomes are naturally secreted from all cell types and bodily fluids, which are enclosed by a phospholipid membrane bilayer. Exosomes are generated in multi-vesicular bodies (MVBs) and secreted when MVBs fuse with the cytoplasmic membrane. In the process, not only some specific transmembrane proteins like CD9, CD63, CD81, CD82 were anchored as a specific marker; the lumen of exosomes also possesses many biological contents, such as soluble enzymes, mitochondrial DNA, mRNA, microRNA, or other functional genes. Their inherent merits to transfer cargos have improved their possibility of being used to deliver a variety of agents, including chemotherapeutics, miRNAs, siRNAs, proteins, and even nanoparticles (*Nat Cell Biol.* 2007, 9(6), 654-659; *Adv Drug Delivery Rev.* 2016, 106, 148-156). Nevertheless, the inherent substances in the lumen have their unique function that may increase the complexity of components for therapeutic application (*ACS Nano.* 2017, 11(1), 69-83; *Cancer Res.* 2003, 63(15), 4331-4337).

Compared with the secreted exosomes, the cytomembrane vesicles possess similar morphological structures, physicochemical properties, and membrane protein components but contain relatively smaller amounts of cellular RNA and other soluble proteins (*Proc Natl Acad Sci.* 2015, 112(45), E6129-E6138). In addition, the cytomembrane vesicles produced from the reassembled cell membrane fraction exhibit high productivity to meet clinical needs (*ACS Nano.* 2013, 7(9), 7698-7710). Therefore, the cytomembrane vesicles may provide an alternative choice for exogenous substances or functional moieties delivery (Fig. R2).

Figure 1b. Schematic describing the production of biNV-IL-15.

Figure R2. The procedure for purification and preparation of cytomembrane vesicles and exosomes (*Adv Sci.* 2021, 8(21), 2100460). The prepared cytomembrane vesicles are usually generated with the addition of ultrasonication to control their size distribution. Benefiting from the complete ultrasonic crushing of primary cells and following a centrifugal separation, cytomembrane vesicles can

minimize the aqueous impurities (e.g., enzymes, oligonucleotides) in the cavity and have great significance in exogenous substances transportation. Exosomes are secreted from all cell types when the multi-vesicular bodies (MVBs) fuse with the cytoplasmic membrane. They play an integral role as intercellular communication vectors because of many soluble proteins and an array of oligonucleotides in the lumen. Besides, they are always regarded as diagnostic biomarkers and therapeutic targets.

3. In the results description for tumor immune microenvironment analysis in Fig. 4C, the authors suggested that MIST treatment converted M2 to M1 macrophages, but in Fig. 5F and S5, S9, the infiltration number of F4/80⁺ cells increased as well as CD8⁺ cells. F4/80 is generally considered a pan marker for macrophage, why would the MIST treatment increase the total number of tumor-infiltrating macrophages?

Response: We appreciate the reviewer's comments. Derived from blood monocytes, macrophages can be recruited by chemokines produced in the tumor microenvironment and therefore infiltrate tumor sites. Among those chemokines, CC chemokine ligand 2 (CCL2) is shown not only to be the most potent chemoattractant factor for monocytes but also to be able to activate them for tumor cell killing (*Cell Mol Immunol.* 2018, 15(4), 335–345; *Cancer Lett.* 2021, 499, 148-163). Many studies have revealed that IL-15 transpresentation could upregulate the production of CCL2 (*J Exp Med.* 2006, 203(10), 2329-2338; *J Immunol.* 2008, 180(10), 6477-6483; *J Immunother Cancer.* 2020, 8(1), e000599), which participates in macrophage recruitment into the tumor site. To explain why the MIST treatment could increase the total number of tumor-infiltrating macrophages, we investigated the levels of CCL2 in tumor tissues through immunohistochemical staining. As shown in Fig. R3-5, biNV-IL-15 treatment significantly elevated the intratumoral CCL2 level in the 4T1 tumor model, CT26 tumor model, and tumor resection model, leading to enhanced tumor infiltration of macrophages.

Figure R3. Immunohistochemical staining of tumors displaying CCL2 levels for PBS and biNV-IL-

15 groups in 4T1 tumor model. Scale bar = 100 μ m.

Figure R4. Immunohistochemical staining of tumors displaying CCL2 levels for PBS and biNV-IL-15 groups in CT26 tumor model. Scale bar = 100 μ m.

Figure R5. Immunohistochemical staining of tumors displaying CCL2 levels for PBS and biNV-IL-15 groups in tumor resection model. Scale bar = 100 μ m.

4. The authors used recombinant IL15 as a control group. In the method part, the authors only mentioned that the recombinant IL15 was from Abcam Inc. without Cat # information. Is the recombinant IL15 is the monomer cytokine only, or the IL15-IL15Ralpha heterodimer complex as the authors expressed in dendritic cells? This is important, as the latter can better serve as a control group for the MIST treatment group. For example, NIZ985, a recombinant heterodimer of IL15-IL15Ralpha, are currently under clinical trials for cancer therapy. So, in my opinion the MIST platform, but not the transpresentation, is the major point of this manuscript.

Response: We appreciate the reviewer's comments. The recombinant IL-15 is the monomer cytokine. As reviewer suggested, we constructed a heterodimer cytokine (IL-15:IL-15R α , also known as hetIL-15/NIZ985) consisting of IL-15 and IL-15R α as a control group for the MIST treatment

group, which was produced and purified from a cloned cell line derived from the HEK293 cell line transfected with optimized plasmid DNA encoding IL-15 and IL-15R α (*J Immunother Cancer*. 2020, 8(1), e000599; *J Biol Chem*. 2013, 288(25), 18093-18103). We then compared the therapeutic outcomes of IL-15:IL-15R α , IL-15:IL-15R α +biNV, and biNV-IL-15 in the 4T1 tumor model, tumor resection model, and hematogenous metastasis model. In contrast to IL-15:IL-15R α and IL-15:IL-15R α +biNV, biNV-IL-15 exhibited superior antitumor activity, anti-recurrence effects, and metastasis-inhibitory ability, demonstrating the promise for practical clinical application. The immune responses after various treatments were also explored through flow cytometric assays. The results indicated that biNV-IL-15 treatment effectively activated antitumor immunity including improved proportion of CD8⁺ T cells, M1-type macrophage polarization, and increased T_{EM} percentage, demonstrating significant advantages over IL-15:IL-15R α and IL-15:IL-15R α +biNV.

The results were discussed in the revised manuscript : “Furthermore, the 4T1 tumor model was also established to compare the therapeutic outcomes of heterodimeric IL-15 (IL-15:IL-15R α) and the MIST platform (Supplementary Fig. 9a). As shown in Supplementary Fig. 9b, the biNV-IL-15 exhibited superior antitumor activity compared with IL-15:IL-15R α and IL-15:IL-15R α +biNV, leading to prolonged survival period of mice”; “The flow cytometric assays were also performed in 4T1 tumor model established to compare the efficacy of heterodimeric IL-15 and the MIST platform. As shown in Supplementary Fig. 9c, e, biNV-IL-15 effectively enhanced the proportion of CD8⁺ T cells and decreased the immunosuppressive Tregs in tumors and dLNs, displaying obvious advantages over IL-15:IL-15R α and IL-15:IL-15R α +biNV. Meanwhile, biNV-IL-15 significantly remodeled TAMs from M2 type to M1 type and upregulated T_{EM} level in the spleen in comparison with IL-15:IL-15R α and IL-15:IL-15R α +biNV (Supplementary Fig. 9c, d)”; “Furthermore, the tumor resection model was also established to compare the anti-recurrence effects of heterodimeric IL-15 and the MIST platform (Supplementary Fig. 16a). As shown in Supplementary Fig. 16b-d, the biNV-IL-15 displayed the excellent capability of tumor recurrence inhibition in contrast to IL-15:IL-15R α and IL-15:IL-15R α +biNV, resulting in superior survival outcomes”; “The flow cytometric assays were also performed in tumor resection model established to compare the efficacy of heterodimeric IL-15 and the MIST platform. As shown in Supplementary Fig. 16e-h, biNV-IL-15 obviously improved the percentage of tumor-infiltrating CD8⁺ T cells and downregulated the level of immunosuppressive Tregs in tumor, implying remarkable advantages over IL-15:IL-15R α and IL-15:IL-15R α +biNV. Furthermore, biNV-IL-15 effectively polarized M2-like TAMs to M1-like TAMs and increased splenic T_{EM} proportion in contrast to IL-15:IL-15R α and IL-15:IL-15R α +biNV (Supplementary Fig. 16i-n)”; “Moreover, the hematogenous metastasis model was also established to compare the antimetastatic ability of heterodimeric IL-15 and the MIST platform (Supplementary Fig.

19a). As shown in Supplementary Fig. 19b-k, the biNV-IL-15 treatment not only inhibited the tumor growth but also prevented lung and liver metastasis of the tumor in contrast to IL-15:IL-15R α and IL-15:IL-15R α +biNV, causing an extended survival time”; “The flow cytometric assays were also performed in hematogenous metastasis model established to compare the metastasis-inhibitory ability of heterodimeric IL-15 and the MIST platform. As shown in Supplementary Fig. 19l-o, biNV-IL-15 significantly elevated the level of CD8⁺ T cells and decreased the proportion of Tregs in blood, demonstrating distinctive advantages over IL-15:IL-15R α and IL-15:IL-15R α +biNV”.

Supplementary Fig. 9. *In vivo* antitumor activity of biomimetic nanovaccine-mediated multivalent IL-15 self-transpresentation (MIST). (a) 4T1 cancer cells were subcutaneously inoculated into BALB/c mice. On days 7, 10, 13, and 16, the PBS, IL-15:IL-15R α , IL-15:IL-15R α +biNV, and biNV-IL-15 were intravenously administered into the mice. (b) Average tumor growth curve and survival curve following various treatments (n = 6/group). (c) Flow cytometric quantification of tumor-infiltrating CD8⁺ T cells, CD4⁺Foxp3⁺ Tregs, M1-type TAMs (CD80^{hi}CD11b⁺F4/80⁺), and M2-type TAMs (CD206^{hi}CD11b⁺F4/80⁺) in 4T1 tumor model following various treatments (n = 4/group). (d) Flow cytometric quantification of CD3⁺CD8⁺CD62L^{low}CD44^{hi} effector memory T cells (T_{EM}) in the spleen after various treatments (n = 4/group). (e) Flow cytometric quantification of CD8⁺ T cells and CD4⁺Foxp3⁺ Tregs in dLNs (n = 4/group). Data represent the mean \pm s.d. Statistical significance was calculated through two-tailed student’s t-test (b), log-rank (Mantel-Cox) test (b), or one-way ANOVA using a Tukey post-hoc test (c-e).

Supplementary Fig. 16. The suppression of tumor recurrence. (a) Schematic depicting biNV-IL-15 treatment in 4T1-luc orthotopic cancer model with incomplete cancer resection (IVIS: *in vivo* imaging system; FC: flow cytometry analysis). (b) *In vivo* bioluminescence imaging for 4T1-luc tumor following the removal of the primary tumor. Every group showed four representative mice. Images on day 10 were shown before surgery. (c) Average tumor growth curve and (d) survival curve (n = 6/group) in tumor resection model receiving various treatments. (e) Flow cytometric examination images and (f) relative quantification of CD8⁺ T cell in tumor (n = 4/group). (g) Flow cytometric measurement images and (h) relative quantification of CD4⁺Foxp3⁺ Tregs in tumor (n = 4/group). (i) Flow cytometric assessment images and (j) relative quantification of M1-type TAMs (CD80^{hi}CD11b⁺F4/80⁺) in tumor (n = 4/group). (k) Flow cytometric analysis images and (l) relative quantification of M2-type TAM (CD206^{hi}CD11b⁺F4/80⁺) in tumor (n = 4/group). (m) Flow cytometric evaluation images and (n) relative quantification of CD3⁺CD8⁺CD62L^{low}CD44^{hi} T_{EM} in the spleen (n = 4/group). Data represent the mean ± s.d. Statistical significance was calculated through two-tailed student's t-test (c), log-rank (Mantel-Cox) test (d), or one-way ANOVA using a Tukey post-hoc test (f, h, j, l, n).

Supplementary Fig. 19. *In vivo* anti-metastasis performance. (a) 4T1-luc cancer cells were intravenously injected into the tumor-bearing mice on day 10 to simulate the hematogenous metastasis model (FC: flow cytometry analysis; H&E staining: hematoxylin and eosin staining). PBS, IL-15:IL-15R α , IL-15:IL-15R α +biNV, and biNV-IL-15 were individually administered on days 7, 10, 13, and 16. (b) *Ex vivo* bioluminescent imaging and (c) average bioluminescent radiance for isolated lungs were studied after various treatments on day 20. (d) Average tumor growth curve and (e) survival curve (n = 6/group) in hematogenous metastasis model receiving various treatments. (f) Photographs and (g) quantification of lung metastases (n = 5/group) for Bouin's fluid staining of lungs. H&E staining of the lung (h) and liver (j) slices after different treatments. Scale bar = 100 μ m. Quantification for metastasis area percentages of the lung (i) and liver (k) slices (n = 3/group). (l) Flow cytometric analysis images and (m) relative quantification of CD8⁺ T cells in the blood (n = 4/group). (n) Flow cytometric examination images and (o) relative quantification of CD4⁺Foxp3⁺ Tregs in the blood (n = 4/group). Data represent the mean \pm s.d. Statistical significance was calculated through one-way ANOVA using a Tukey post-hoc test (c, g, i, k, m, o), two-tailed student's t-test (d) or log-rank (Mantel-Cox) test (e).

5. Minor points: In the introduction part, page 4, line 56-58, the authors wrote: "IL-15 supports the generation and survival of memory T cells, while IL-2 promotes immunosuppressive regulatory T cell (Treg) expansion". This is misleading, as IL2 is a T cell survival factor for all T cell subsets including effector T cells and Tregs. In another sentence, the authors claimed "In addition, the distinct size of IL-15 is below the renal filtration threshold (<40 kDa), resulting in rapid clearance and short-lived effects." This is partially true, but it is not the main reason for the limit of IL15 clinical use. For example, IFNalpha and IL2 are also small proteins, but both have been approved by FDA for cancer treatment. Please revise these sentences to accurately describe the current limit of clinical use of cytokines in cancer therapy.

Response: We appreciate the reviewer's comments. As reviewer suggested, we revised these sentences in the introduction part: "IL-15 supports the generation and survival of memory T cells, while IL-2 promotes the expansion of all T cell subsets including effector T cells and immunosuppressive regulatory T cells (Tregs)"; "The weak interaction with tumor-specific T cells of IL-15 raises the risks of nontumor-specific immune activation, leading to systemic side effects and a deficiency of tumor suppression".

Reviewers' Comments:

Reviewer #1:

Remarks to the Author:

The authors have adequately addressed my concerns.

Reviewer #2:

Remarks to the Author:

The Authors of the manuscript NCOMMS-23-00992A Biomimetic nanovaccine-mediated multivalent IL-15 self-transpresentation (MIST) for potent and safe cancer immunotherapy have accomplished an accurate revision of their article answering to the addressed remarks and suggestions. The text results upgraded and the manuscript appears suitable for publication.

Reviewer #3:

Remarks to the Author:

The authors have addressed all my questions, and the manuscript has been significantly improved. I would recommend acceptance of the revised manuscript for publication.